# LAYERFUSION: HARMONIZED MULTI-LAYER TEXT-TO-IMAGE GENERATION WITH GENERATIVE PRIORS

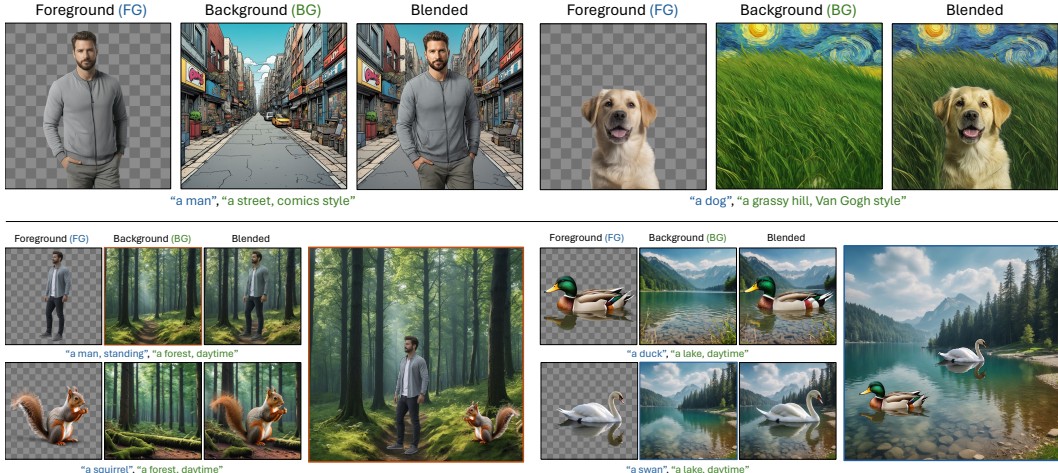

Figure 1: **LayerFusion.** We propose a framework for generating a foreground (RGBA), background (RGB) and blended (RGB) image simultaneously from an input text prompt. By introducing an optimization-free blending approach that targets the attention layers, we introduce an interaction mechanism between the image layers (i.e., foreground and background) to achieve harmonization during blending. Furthermore, as our framework benefits from the layered representations, it enables performing spatial editing with the generated image layers in a straight-forward manner.

## ABSTRACT

Large-scale diffusion models have achieved remarkable success in generating high-quality images from textual descriptions, gaining popularity across various applications. However, the generation of layered content, such as transparent images with foreground and background layers, remains an under-explored area. Layered content generation is crucial for creative workflows in fields like graphic design, animation, and digital art, where layer-based approaches are fundamental for flexible editing and composition. In this paper, we propose a novel image generation pipeline based on Latent Diffusion Models (LDMs) that generates images with two layers: a foreground layer (RGBA) with transparency information and a background layer (RGB). Unlike existing methods that generate these layers sequentially, our approach introduces a harmonized generation mechanism that enables dynamic interactions between the layers for more coherent outputs. We demonstrate the effectiveness of our method through extensive qualitative and quantitative experiments, showing significant improvements in visual coherence, image quality, and layer consistency compared to baseline methods.

## 1 INTRODUCTION

Large-scale diffusion models have recently emerged as powerful tools in the domain of generative AI, achieving remarkable success in generating high-quality, diverse, and realistic images from textual prompts. These models, such as DALL-E (Ramesh et al. (2021)), Stable Diffusion (Rombach

et al. (2022)), and Imagen (Saharia et al. (2022)), have gained immense popularity due to their ability to create complex visual content with impressive fidelity and versatility. As a result, they have become integral to various applications, from digital art and entertainment to data augmentation and scientific visualization. However, despite the significant advancements in these models, the problem of layered content generation has only recently been explored by works such as Zhang et al. (2024); Quattrini et al. (2024), which exhibits the potential of enabling creative workflows.

Layered content generation, especially the creation of transparent images, plays a vital role in creative industries such as graphic design, animation, video editing, and digital art. These workflows are predominantly layer-based, where different visual elements are composed and manipulated on separate layers to achieve the desired artistic effects. Transparent image generation, where the foreground content is isolated with an alpha channel (RGBA), is essential for blending different visual elements seamlessly, enhancing flexibility, and ensuring coherence in complex visual compositions. The lack of research on generating such layered content highlights an important gap considering the application of diffusion models in practical and creative context, in addition to the usefulness of layer-based content creation tools such as Adobe Photoshop and Canva.

To address this gap, we propose a novel image generation pipeline based on Latent Diffusion Models (LDMs) (Rombach et al. (2022)) that focuses on generating layered content. Our method produces images with two distinct layers: a foreground layer in RGBA format, containing transparency information, and a background layer in RGB format. This approach contrasts with traditional methods that are either based on generating layered content in a sequential manner (Quattrini et al. (2024)), or empowered by sequentially generated synthetic data in less satisfying quality (Zhang et al. (2024)), which often leads to inconsistencies and lack of harmony between generated layers. Instead, our proposed method introduces a harmonized generation mechanism that enables interactions between these two layers, resulting in more coherent and appealing outputs and supporting flexible spatial edits for manipulation, as shown in Fig. 1.

In our framework, the harmonization between foreground and background layers is achieved through the utilization of cross-attention and self-attention masks extracted from the foreground generation model. These masks play a critical role in guiding the generation process by identifying and focusing on the relevant features needed to create both layers in a unified manner. By leveraging these attention mechanisms, our approach allows for a fine-grained control over the generation process, ensuring that the generated foreground and background elements interact naturally, enhancing the overall visual quality and coherence. Following this, we introduce an innovative attention-level blending mechanism that utilizes the extracted attention masks to generate the background and foreground pair in a harmonized manner. Unlike previous methods that handle each layer separately and rely on training data that involves sequentially generated layers (Zhang et al. (2024)), our blending scheme integrates information from both layers at the attention level, allowing for dynamic interactions and adjustments that reflect the underlying relationships between the elements of the scene. This not only improves the realism of the generated images but also provides users with enhanced control over the final composition. In summary, our contributions are threefold:

- We propose a new image generation pipeline that generates images with two layers—foreground (RGBA) and background (RGB)—in a harmonized manner, allowing for natural interactions between the layers.

- We develop a novel attention-level blending scheme that uses the extracted masks to perform seamless blending between the foreground and background layers. This mechanism ensures that the two layers interact cohesively, leading to more natural and aesthetically pleasing compositions.

- We perform extensive qualitative and quantitative experiments that demonstrate the effectiveness of our method in generating high-quality, harmonized layered images. Our approach outperforms baseline methods in terms of visual coherence, image quality, and layer consistency across several evaluation metrics.

## 2 RELATED WORK

**Denoising Probabilistic Diffusion Models.** Diffusion models contributed significantly in the field of image generation, specifically for the task of text-to-image generation. In early efforts, Ho et al.

(2020); Song et al. (2020a;b) made significant contributions to the area, where significant improvements on generation performance has been experienced with diffusion models on pixel level. In another paradigm Rombach et al. (2022) proposed operating in a latent space, which enabled the generation of high-quality images with a lower computation cost compared to models operating on pixel-level, which built the foundation of the state-of-the-art image generation models Podell et al. (2023); Esser et al. (2023); Peebles & Xie (2023). Even though such approaches differ in terms of their architecture designs, they all follow a paradigm that prioritizes building blocks relying on attention blocks Vaswani (2017).

**Transparent Image Layer Processing.** In terms of obtaining single foreground layer, the work of Chen et al. (2022) presents PP-Matting, a trimap-free natural image matting method that achieves high accuracy without requiring auxiliary inputs like user-supplied trimaps. Meanwhile, Quattrini et al. (2024) propose Alfie, a method for generating high-quality RGBA images using a pre-trained Diffusion Transformer model, designed to provide fully-automated, prompt-driven illustrations for seamless integration into design projects or artistic scenes. It modifies the inference-time behavior of a diffusion model to ensure that the generated subjects are centered and fully contained without sharp cropping. It utilizes cross-attention and self-attention maps to estimate the alpha channel, enhancing the flexibility of integrating generated illustrations into complex scenes. In terms of multi-layer, Tudosiu et al. (2024) recently introduce MuLAn, a novel dataset comprising over 44,000 multi-layer RGBA decompositions of RGB images, designed to provide a resource for controllable text-to-image generation. MuLAn is constructed using a training-free pipeline that decomposes a monocular RGB image into a stack of RGBA layers, including background and isolated instances. While these methods have made significant progress, precise control over image layers and their harmonization remain challenging.

The most related effort for layered content synthesis is done by Zhang et al. (2024). This approach is notable for its capability to generate both single and multiple transparent image layers with minimal alteration to the original latent space of a pretrained diffusion model. The method utilizes a "latent transparency" that encodes the alpha channel transparency into the latent manifold of the model. It offers two main workflows. One is jointly generating foreground and background layers by attention sharing. The other one is a sequential approach that generates one layer first and then another layer based on previous layer. Both requires heavy model training relying on synthetic training data in less satisfying quality (obtained by a pretrained inpainting model). In contrast, our framework provides a training-free solution that offers generation of layered content in a simultaneous manner, which both benefits from layer transparency and achieves harmony between layers.

## 3 METHOD

### 3.1 THE LAYERDIFFUSE FRAMEWORK

For the foreground generation, we rely on the LayerDiffuse framework proposed by Zhang et al. (2024). As a preliminary step to achieve foreground transparency, it initially introduces a latent transparency offset $x_\epsilon$, which adjusts the latents $x$ decoded by the VAE of the latent diffusion model, to obtain a latent distribution modelling foreground objects as $x_a = x + x_\epsilon$. Following this step, they train a transparent VAE $D(\hat{I}, x_a)$, that predicts the $\alpha$ channel of the RGB image involving a single foreground image, which is referred to as the pre-multiplied image $\hat{I}$. Note that our framework only benefits from their foreground generation model, proposing a training-free solution of generating blended and background images without needing any additional training, without disturbing the output distribution of neither the foreground or the original pretrained diffusion model. A visual overview of our framework is provided in Fig. 2.

### 3.2 ATTENTION MASKS AS GENERATIVE PRIORS

To perform harmonized foreground and background generation, we introduce a blending scheme that focuses on combining attention outputs with a mask that provides sufficient information about both the content and the structure of the foreground latent being diffused. To achieve this task, we utilize self-attention and cross-attention probability maps of the foreground generator as structure and content priors for the generative process, respectively. Note that each of these probability maps

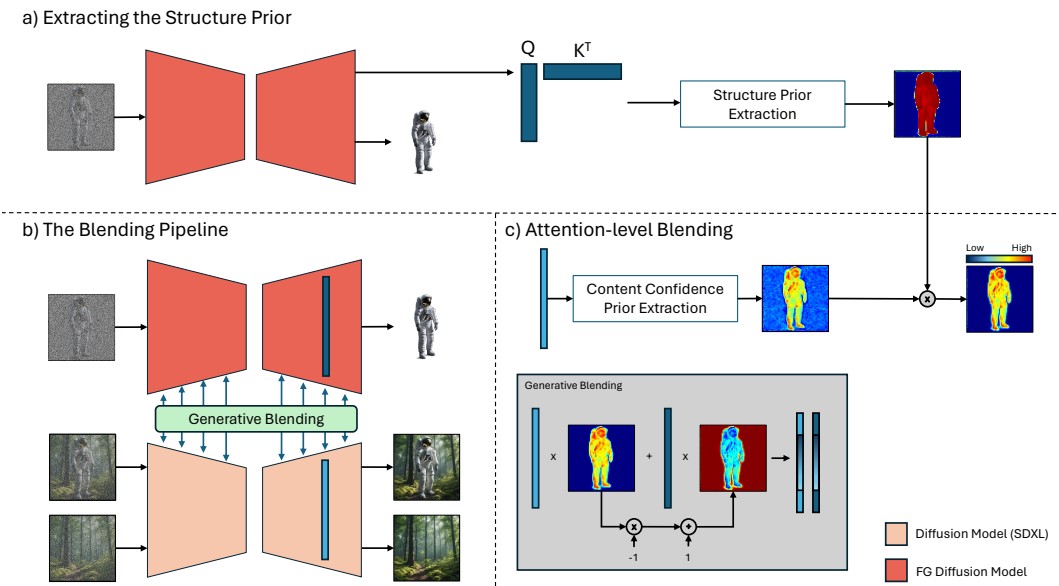

Figure 2: **LayerFusion Framework.** By making use of the generative priors extracted from transparent generation model $\epsilon_{\theta,FG}$, LayerFusion is able to generate image triplets consisting a foreground (RGBA), a background, and a blended image. Our framework involves three fundamental components that are connected with each other. First we introduce a prior pass on $\epsilon_{\theta,FG}$ (a) for extracting the structure prior, and then introduce an attention-level interaction between two denoising networks ($\epsilon_{\theta,FG}$ and $\epsilon_{\theta}$) (b), with an attention level blending scheme with layer-wise content confidence prior, combined with the structure prior (c).

are formulated as $softmax(\frac{Q \cdot K^T}{\sqrt{d}})$ where $Q$ and $K$ are the query and key features of the respective attention layer. Below, we explain how such attention masks are getting extracted in detail for both structure and content related information.

**Extracting Structure Prior.** During the blending process, we bound the blending region with a structure prior extracted from the foreground generation model, $\epsilon_{\theta,FG}$. To extract a boundary for the foreground generated by $\epsilon_{\theta,FG}$, we utilize the attention probability map $m \in \mathbb{R}^{MxM}$ of the corresponding self-attention layer, averaged over its attention heads. Upon investigating what each of these values correspond to, we interpret that the last dimension of the probability map implies a probability distribution of the cross correlation values between a variable and all of the other variables processed by the self-attention layer, where $M$ is the number of variables processed by each attention block.

Furthermore, since the foreground model $\epsilon_{\theta,FG}$ is trained specifically for generating a single subject as the foreground object, we interpret the density of the distribution of the cross-correlation values of a variable as a vote on whether that variable is a foreground or not. To quantify this observation, we introduce a per-variable sparsity score $s_i = \frac{1}{\sum_{j=1}^{M} m_{i,j}^2}$ where $s_i$ is the sparsity score for variable $i$, followed by a min-max normalization. Since the estimate $s_i$ measures how sparse the cross-correlation value distribution of the variable $i$, we negate these values to get a density estimate by $s_i' = 1 - normalize(s_i)$, favoring dense probability distributions over sparse ones.

Given the formulation of the sparsity estimates $s_i$ for variable $i$, we capture the structure information the best on the preceding layers of the foreground diffusion model $\epsilon_{\theta,FG}$. To capture the structure prior, we utilize the last self attention layer of the diffusion model where we provide additional analyses in supplementary material.

**Retrieving Content Confidence Priors.** As the second component of our blending scheme, we extract content confidence priors as attention maps to be able to blend background and foreground in a seamless manner. To do so, we utilize cross-attention maps of the transformer layer, where blending operation occurs. Utilizing the unidirectional nature of CLIP Text Encoder, we extract

Figure 3: **Visualization of the masks extracted as generative priors.** Throughout the generation process, we extract a structure prior $s$ and a content confidence prior $c$. To combine the structure and content information, we construct $mask_{soft}$ and $mask_{hard}$ during the blending process. As visible from the provided maps (as priors), We can both capture the overall object structure with the structure prior $s$ and incorporate the content with $c$, where their combination provides a precise mask reflecting both quantities (see the example "the car"). Also note that the masks we construct also capture transparency information throughout the masking process (see the example "a glass bottle"). We retrieve the provided masks for the diffusion timestep $t = 0.8T$.

the content confidence map from `<EOS>` attention probability map, to accumulate all information related to the foreground, following the observations presented in Yesiltepe et al. (2024). Similar to extracting the structure prior, we again use $\epsilon_{\theta, FG}$ for extracting the foreground related information, benefiting from the fact that the model is conditioned on generating a single object, which is the foreground object itself.

Among the cross-attention probabilities, we utilize the cross-attention probability values $n \in \mathbb{R}^{HxMxT}$ of the conditional estimate, conditioned by the foreground prompt where the cross-attention layer has $H$ heads, and $T$ is the number of text tokens inputted. Using these probability maps, we extract a soft content confidence map $c$ to quantify how much of an influence does the input condition (prompt) has on the generated foreground. To do so, we utilize the mean of the cross-attention probability maps over $H$ attention heads.

## 3.3 BLENDING SCHEME

Given the formulations for the structure prior and the content confidence maps, extracted from the foreground generator, $\epsilon_{\theta, FG}$, we propose a blending scheme on the attention level to achieve full harmonization. Since the content of the generated image is constructed gradually in every consecutive attention layer, where self-attention focuses more on the structure details and cross-attention focuses more on the content of the image, we introduce a blending scheme that targets both, with the help of generative priors extracted from these targeted layers. For the blending scheme, we first introduce a mask extraction algorithm where we extract soft and hard blending masks for the given attention block. Given the structure prior $s$ and content confidence prior $c$, we initially extract $mask_{soft}$ as $s * c$, followed by a min-max normalization to be able to use it as a blending mask. Then, to identify the regions that are affected by the soft blending, we extract our hard mask $mask_{hard}$ by using the soft decision boundary $\sigma(d * (mask_{soft} - 0.5))$, where $\sigma$ is the sigmoid operator. During blending, we select the decision boundary coefficient $d$ as 10, where we provide ablations in Sec. 4.1.4. We provide visualizations of prior masks $s, c$ and blending masks $mask_{soft}$ and $mask_{hard}$ in Fig. 3.

After extracting the soft and hard blending masks, we perform blending in attention level. Given an image generation procedure where one aims to generate an image triplet consisting a foreground, background and blended image, we introduce a blending approach involving the attention outputs of the blended image, $a_{Blended}$, and the foreground image, $a_{FG}$. As the soft mask $mask_{soft}$ encodes the structure and content information related with the foreground, we initially perform soft attention blending between the blended and foreground attention outputs, to reflect the foreground content on the blended image. We formulate the blending equation in Eq. 1.

$$a'_{Blended} = a_{FG} * mask_{soft} + a_{Blended} * (1 - mask_{soft}) \tag{1}$$

Following this initial blending step, we update the attention output for the foreground image with the blending result with hard mask $mask_{hard}$, which is formulated as Eq. 2. This way, we both enable

consistency across the blended image and the foreground image, and enable information transfer between the RGB image generator $\epsilon_\theta$ and foreground image generator $\epsilon_{\theta,FG}$.

$$a'_{FG} = a'_{Blended} * mask_{hard} + a_{FG} * (1 - mask_{hard}) \tag{2}$$

As the final component of generating the desired triplet, we introduce an attention sharing mechanism between the blended hidden states, $h_{blended}$, and background hidden states, $h_{BG}$, to encourage generating a background consistent with the blended image for both the self-attention and cross-attention blocks. We formulate the full blending algorithm in the supplementary material. Note that our approach uses $a'_{Blended}, a'_{FG}$ and $a_{BG}$ as the attention outputs.

# 4 EXPERIMENTS

In all of our experiments, we use SDXL model as the diffusion model. Following the implementation released by Zhang et al. (2024), we use the model checkpoint `RealVisXL_V4.0`[1], unless otherwise stated. While using the non-finetuned SDXL, $\epsilon_\theta$ as the background and blended image generators, we use the weights released by Zhang et al. (2024) for the foreground diffusion model $\epsilon_{\theta,FG}$[2]. We conduct all of our experiments on a single NVIDIA L40 GPU.

## 4.1 QUALITATIVE RESULTS

### 4.1.1 COMPARISONS WITH LAYERED GENERATION METHODS

We compare our proposed method against LayerDiffuse to evaluate the quality of the generated foreground (FG), background (BG), and the blended image (see Fig. 5). As shown in the results, our model achieves harmonious blending with smooth FG and BG images. In contrast, LayerDiffuse (Generation) struggles to produce a smooth and consistent background (see the artifacts in Fig. 5 (b) in the background images). This limitation arises from the sequential approach used to curate the training dataset of LayerDiffuse Zhang et al. (2024), where given a foreground and a blended image, the background is generated by outpainting the foreground from the blended image with SDXL-Inpainting Podell et al. (2023). As a result of this strategy on dataset generation, the background generation model experiences artifacts in the outpainted region, which propagates from the inpainting model. As it is also highlighted in Fig. 5, such artifacts effect the ability of performing spatial edits with the generated foreground and background layers.

### 4.1.2 COMPARISONS WITH FOREGROUND EXTRACTION METHODS

As another baseline, we compare our proposed framework with foreground extraction methods given the blended image (background and blended for LayerDiffuse(Zhang et al. (2024))) to outline the advantages of simultaneous generation of the foreground and background images (layers) in Fig. 6. In addition to background and blended image conditioned foreground extraction pipeline of Zhang et al. (2024), we also consider PPMatting (Chen et al. (2022)) and MattingAnything (Li et al. (2024)) as competitors as they apply matting to extract the foreground layer from the blended image. As we demonstrate qualitatively in Fig 6, simultaneous generation results in more precise foreground for the cases that include interaction between foreground and background layers (e.g. legs of the horse occluded in the grass) compared to state-of-the art foreground extraction/matting methods.

### 4.1.3 COMPARISONS ON HARMONIZATION QUALITY

For the evaluation of the blending capabilities of our framework, we compare our generative blending result with state-of-the-art image harmonization methods. In our comparisons, we investigate the realism of the harmonized output considering the object (foreground) getting harmonized in the process. To get the harmonized outputs from the competing methods, we give the alpha blending result obtained from our pipeline to each of the competitor methods, and qualitatively evaluate the obtained outputs in Fig. 7. Specifically, we compare our framework with Ke et al. (2022); Chen et al. (2023); Guerreiro et al. (2023).

---

[1]`https://huggingface.co/SG161222/RealVisXL_V4.0`
[2]`https://huggingface.co/lllyasviel/LayerDiffuse_Diffusers`

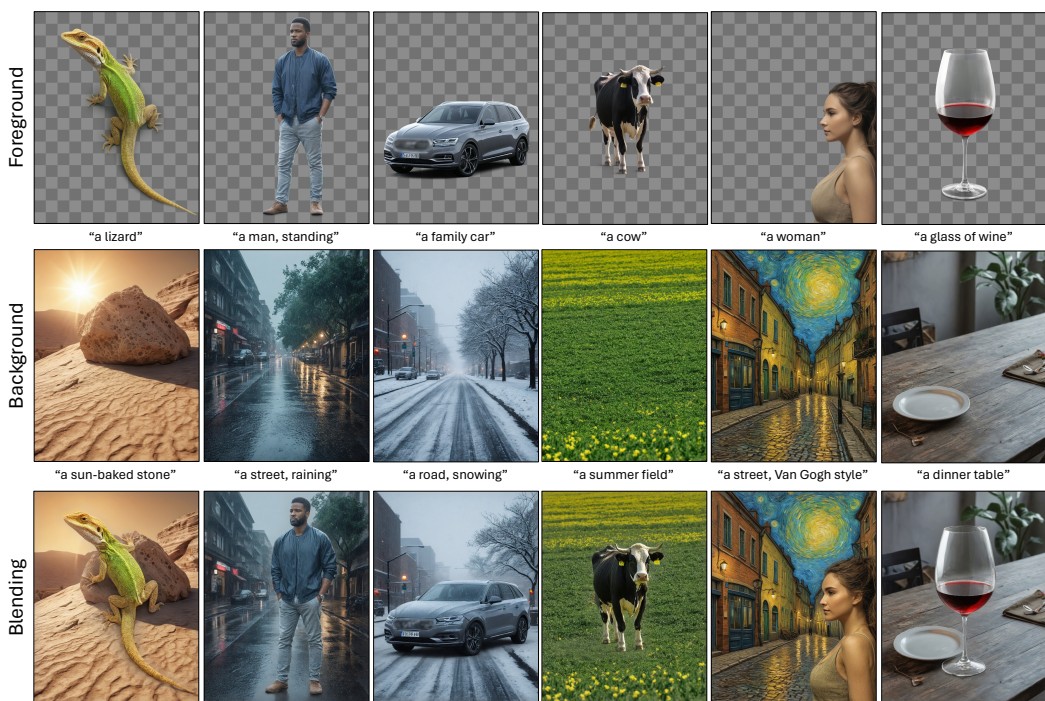

Figure 4: **Qualitative Results.** We present qualitative results on multi-layer generation over different visual concepts. In each column, we show the high-quality results of foreground layer, background layer and their generative blending respectively, in terms of text-image alignment, transparency and harmonization. We present more results in the supplementary material.

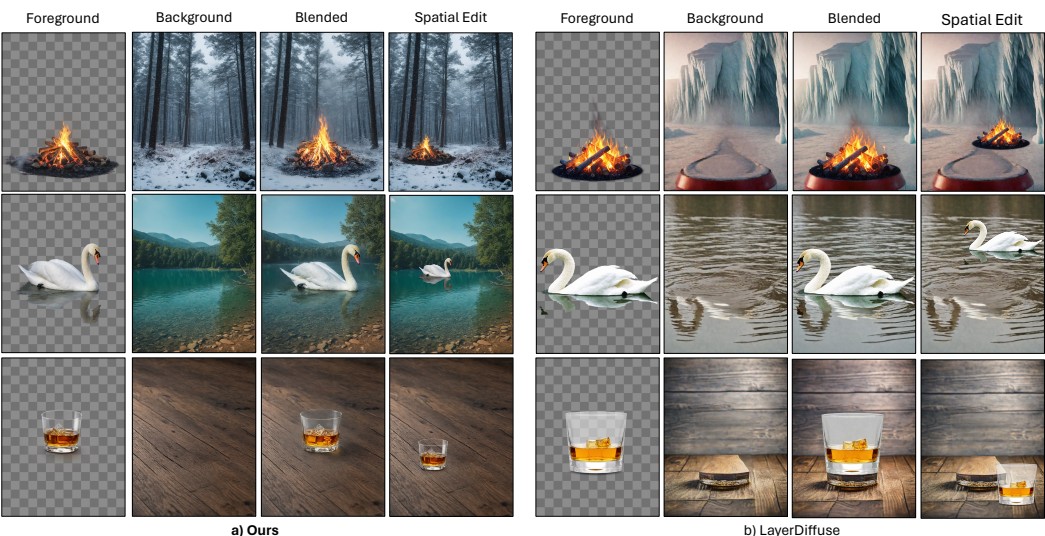

Figure 5: **Qualitative Comparisons on Layered Generation.** We compare our proposed framework with Zhang et al. (2024) to evaluate the performance in terms of layered image generation (e.g. foreground, background, blended). It clearly shows that Zhang et al. (2024) propagates the background completion issues observed in SDXL-Inpainting, which degrades the spatial editing quality with the outputted layers. In constrast, our method can provide both harmonized blending results and isolated foreground and background, which enables spatial editing in a straight-forward manner.

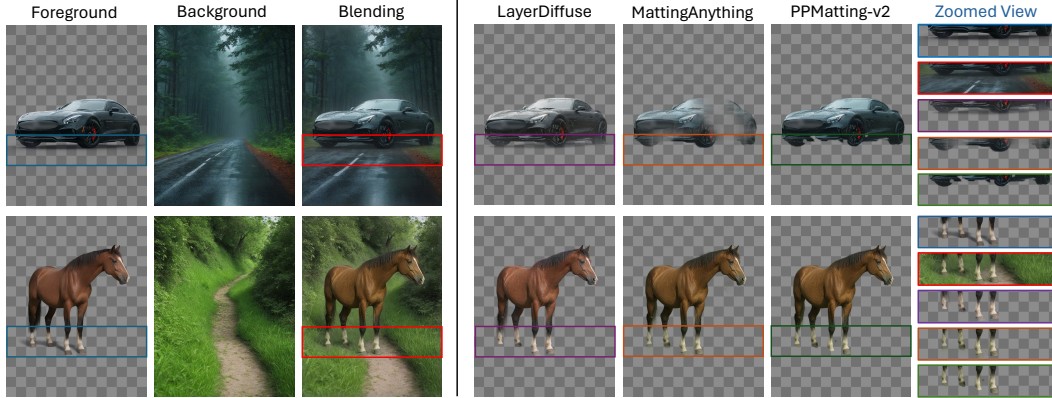

Figure 6: **Comparisons with Foreground Extraction Methods** To illustrate the advantage of our method over the task of foreground extraction given a blended image, we qualitatively compare our approach with LayerDiffuse (Zhang et al. (2024)), Matting Anything (Li et al. (2024)), and PPMatting (Chen et al. (2022)). As also highlighted by Zhang et al. (2024), simultaneous generation of the foreground layer is more advantageous compared to extracting from the blended image in terms of the quality of the foreground image.

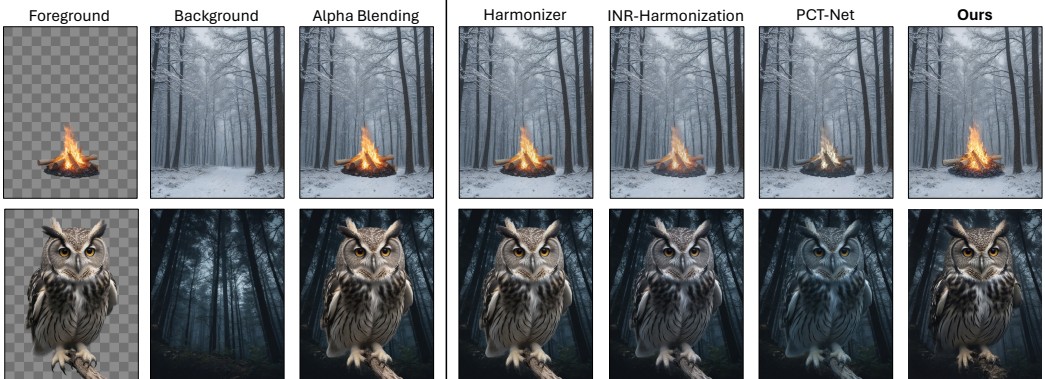

Figure 7: **Comparisons on Image Harmonization.** We qualitatively evaluate our methods blending capabilities by comparing with image harmonization methods Harmonizer (Ke et al. (2022)), INR-Harmonization (Chen et al. (2023)), and PCT-Net (Guerreiro et al. (2023)). Our proposed generative blending approach results in adaptation of the foreground object to the background scene (e.g. snow effect on the campfire), in addition to harmonization methods.

### 4.1.4 ABLATION STUDIES

**Influence of BG on FG.** We explore how changes in the background prompt affect the generated foreground content. As shown in Fig. 8 (a), by varying the background conditions, such as changing weather scenarios, leads to corresponding adjustments in the foreground details, like the clothing or accessories of a person, as well as fine-grained details such as adding snow on the boots (see rightmost image in Fig. 8 (a)). All experiments are conducted using the same seed, allowing for the preservation of the subject's identity while adapting other features to match the changing background context. This demonstrates the dynamic adaptability of our method, where the foreground is influenced by the background for more contextually appropriate outputs.

**Alpha Blending vs. Generative Blending.** We compare two blending strategies: Alpha Blending, which guarantees a complete match between the generated foreground and the blended result, and Generative Blending, which aims for a more realistic composition by considering shadows, lighting, and contextual harmonization. As can be seen from Fig. 8 (b), the Alpha Blending is more deterministic, ensuring that the foreground remains consistent with the original output without considering the interactions between foreground and background. Meanwhile, the Generative Blending

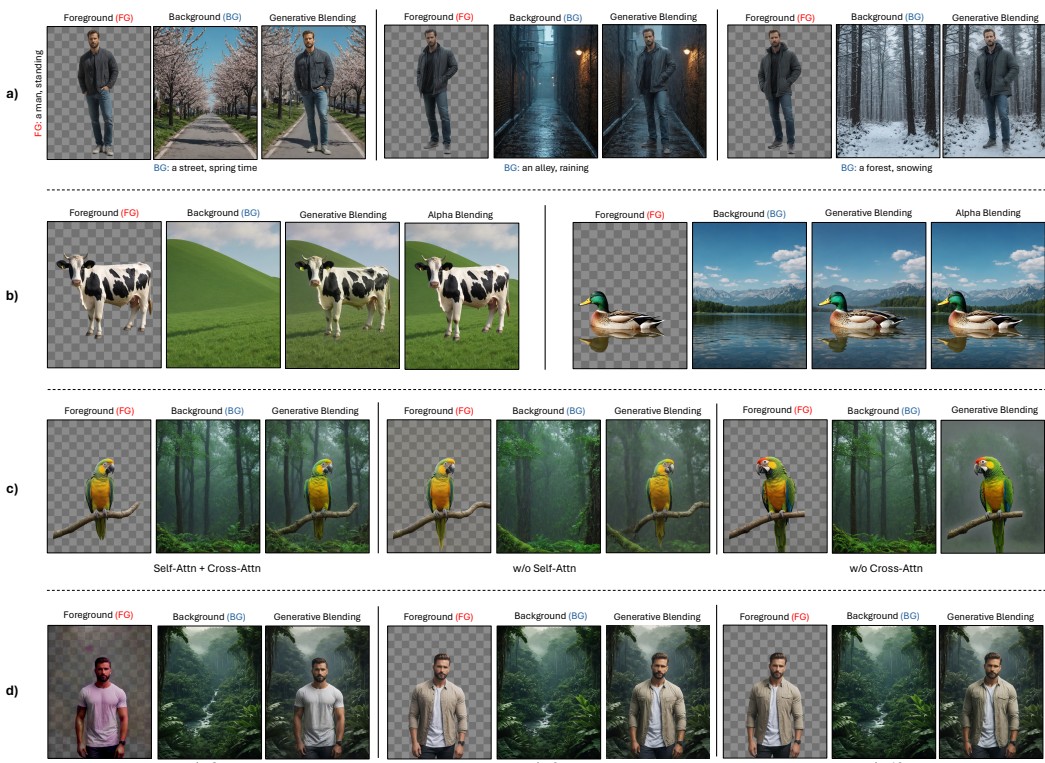

Figure 8: We perform extensive ablation studies on the effect of (a) **Background Influence on Foreground**: Background changes (e.g., weather) dynamically adjust the foreground (e.g., outfit) while preserving identity. (b) **Alpha vs. Generative Blending**: Alpha Blending ensures a perfect match, while Generative Blending creates more realistic harmonization by handling shadows and lighting. (c) **Self-Attention vs. Combined Attention Masks**: Self-attention alone causes leaks; cross-attention alone affects the entire image. Combining both achieves sharper boundaries and better coherence. (d) **Soft Decision Boundary Coefficient**: Lower coefficients cause leaks; higher coefficients yield more precise alpha and consistent blending (e.g., the pocket of the man's clothing).

produces more visually appealing results by better handling subtle elements like shadows and lighting, making the generated content appear more natural and harmonized with the background. Note how the feet of the cow is harmonized with the grassy surface in Generative Blending as opposed to Alpha Blending.

**Self-Attention vs. Cross-Attention.** The use of attention masks plays a crucial role in controlling the interaction between the foreground and background layers. As can be seen from Fig. 8 (c), when the self-attention map is used alone, there are risks of unwanted leaks from the premultiplied image (i.e., the output from the foreground generation model with a gray background), resulting in imprecise boundaries. The cross-attention map, on the other hand, provides more precise information, sharpening the bounding map. However, if the cross-attention map is used in isolation, the regions that are not voted by the structure prior(from the self attention map) may create artefacts. By combining both attention maps, we are able to balance these effects, where the cross-attention sharpens the boundary, and the self-attention ensures coherence within the bounded region.

**Soft Decision Boundary Coefficient.** We investigate the effect of varying the soft decision boundary coefficient, which is used to derive the hard mask in our blending formulation (Eq. 2). Lower coefficients result in softer decision boundaries, causing leaks into the foreground and leading to imprecise alpha channel predictions, as seen in the first image of Fig. 8 (d). As the coefficient value increases, the boundary becomes more defined, allowing for more accurate capture of foreground details and improving consistency between the foreground and blended image. This is particularly evident in the pocket area of the man's clothing in the second and third images, where higher coefficients result in more precise blending and alignment.

| | Foreground | | | Background | | | Blending |
|---|---|---|---|---|---|---|---|
| | CLIP | KID | FID | CLIP | KID | FID | User Preference |
| Zhang et al. (2024) | 38.46 | 0.0014 | 0.09 | 38.27 | 0.0400 | 1.17 | $2.960 \pm 0.692$ |
| Ours | **38.97** | **0.0012** | 0.09 | **41.95** | **0.0058** | **0.14** | **$3.233 \pm 0.566$** |

Table 1: **Quantitative Results.** We quantitatively evaluate the output distribution for the foreground and background images with CLIP-score, KID, and FID metrics. Furthermore, we also conduct a user study to evaluate the blending performance of our framework perceptually.

## 4.2 QUANTITATIVE RESULTS

We compare our framework with Zhang et al. (2024) as the only approach that succeeds in transparent foreground generation, coupled with an RGB background and blending result. Throughout our comparisons, we quantitatively assess the background, foreground and blending quality. Over the presented comparisons, we both demonstrate perceptual evaluations with an user study over the blending results, and analyses over the quality of the generated background and foreground.

**Foreground & Background Quality.** To assess the quality of the backgrounds and foregrounds generated, we first evaluate the prompt alignment capabilities of both approaches over the generated foregrounds and backgrounds with the CLIP score Radford et al. (2021). We use the CLIP-G variant as both text and image encoders throughout our experiments. In addition to prompt alignment properties, we measure how both approaches align with the real imaging distribution for both the foreground and background images. Using the images generated by the foreground generator of Zhang et al. (2024) and backgrounds generated by non-finetuned SDXL as the real imaging distributions, we quantitatively compare our generations in terms of prompt alignment with the CLIP score (Radford et al. (2021)), and the closeness to the real imaging distribution with KID (Bińkowski et al. (2018)) and FID (Heusel et al. (2017)) scores. To evaluate the two image distributions, we use KID score with the final pooling layer features of Inception-V3 (Szegedy et al. (2016)) to evaluate the similarity overall image distribution, and FID score with the features from the first pooling layer to evaluate texture level details. As our results also demonstrate, while preserving the output distribution of the foreground diffusion model, $\epsilon_{\theta,FG}$, our framework offers a background image distribution that aligns better with the RGB diffusion model, $\epsilon_\theta$ (e.g. SDXL).

**User Study.** To perceptually evaluate the quality of the blending performed by our framework, we conduct an user study over the Profilic platform[3], with 50 participants over a set of 40 image triplets. We show the participants the blended image along with the foreground and background images, and ask to rate the blended output over a rate of 1-to-5 (1=not satisfactory, 5=very satisfactory). We present the results of the user study in Table 1 which shows that our results receive higher ratings for more satisfying results. Additional details about the user study setup are provided in the supplementary material.

## 5 LIMITATION AND CONCLUSION

In this paper, we presented a novel image generation pipeline based on LDMs that addresses the challenge of generating layered content, specifically focusing on creating harmonized foreground and background layers. Unlike traditional approaches that rely on consecutive layer generation, our method introduces a harmonized generation process that enables dynamic interactions between the layers, leading to more coherent and aesthetically pleasing outputs. This is achieved by leveraging cross-attention and self-attention masks extracted from the foreground generation model, which guide the generation of both layers in a unified and context-aware manner. It is noted that since our pipeline is built on top of pre-trained LDMs and LayerDiffuse Zhang et al. (2024) which may carry inherent biases from their training data, these biases can affect the generated content, potentially leading to outputs that are not entirely aligned with user expectations or specific requirements. Nevertheless, the findings highlight the potential of our approach to transform creative workflows that rely on layered generation, providing more intuitive and powerful tools for artists and designers.

---

[3]Prolific platform: https://www.prolific.com/

## 6  ETHICS STATEMENT

As addressed in Sec. 5, due to the dependency of our proposed method on pretrained LDMs and the LayerDiffuse Zhang et al. (2024) framework, our method may reflect the biases inherited by the these methods. While encouraging the responsible use of such methods, we also acknowledge the potential biases inherited by these methods. However, as our method does not perform any kind of dataset collection or introduce any new dataset, we leave such issues to the consent of the users. In addition, regarding the user study conducted as a part of our study, we completely acknowledge the anonymity of the participants.

## 7  REPRODUCIBILITY STATEMENT

In order to encourage the reproducibility of the work done in this manuscript, we share the detailed algorithm of the proposed LayerFusion in detail. The proposed approach is not optimization-based, where our method does not include any trained parameters. Furthermore, we facilitate the reproducibility of our approach with a detailed pseudo-code of the proposed approach (see supplementary material for additional details). In addition, for the reproducibility of the provided qualitative results, we provide the input prompts used in order to generate the examples provided along with the public links to the model checkpoints used as a part of this work. In addition, we also provide our experiment setup in detail along with the external networks used for scoring the outputs.

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

# A SUPPLEMENTARY MATERIAL

## A.1 DISCLAIMER

In the provided qualitative results throughout this paper, we apply blurring to any trademark logos visible in the generated samples for copyright issues.

## A.2 LIMITATIONS

While our proposed image generation pipeline based on Latent Diffusion Models (LDMs) demonstrates significant advancements in generating harmonized foreground (RGBA) and background (RGB) layers, there are several limitations that warrant discussion. Our current approach focuses on generating images with two distinct layers—a foreground and a background. While this is suitable for many creative workflows, it does not extend to more complex scenarios involving multiple layers or hierarchical relationships among multiple visual elements, which we intent to explore for future work. Moreover, the harmonization between foreground and background layers in our framework relies heavily on the quality of the cross-attention and self-attention masks extracted from the generation model. In cases where these masks are suboptimal or noisy, the blending of layers may not be as effective, leading to artifacts or less coherent outputs. Finally, our method depends on pre-trained Latent Diffusion Models both for foreground and background generation, which may carry inherent biases from their training data (such as generating centered foregrounds for the RGBA component). These biases can affect the generated content, potentially leading to outputs that are not entirely aligned with user expectations or specific requirements in diverse applications. Nevertheless, our method provides a structured framework for generating transparent images and layered compositions, which are crucial for many creative tasks.

## A.3 ANALYSES ON STRUCTURE PRIORS FROM DIFFERENT LAYERS

In all of the experiments we provide, we utilize the structure prior extracted from the last attention map of the foreground diffusion model, $\epsilon_{\theta,FG}$. As a justification of this decision and to clearly illustrate what different self attention layers focus on throughout the generation process, we provide structure priors extracted from different layers in Fig. 9. As it can also be observed visually, the structure prior extracted from the last self attention layer provides a more precise estimate of the shape of the foreground being generated.

## A.4 DETAILED BLENDING ALGORITHM

Supplementary to the definition of the blending algorithm provided in Sec. 3.3, we provide a more detailed description in this section, for clarity. Our proposed blending approach involves three sub-procedures, which are the extraction of the structure prior, extraction of the content confidence prior and the attention blending step. In this section, we provide the pseudo-code for all three procedures as Alg. 1, 2 and 3.

## A.5 USER STUDY DETAILS

We conduct our user study over 50 participants with 40 image triplets generated by LayerFusion and Zhang et al. (2024). For the generation of the subjected triplets, we generate examples with animal, vehicle, matte objects, person and objects with transparency properties as the foreground to get samples representing a diverse distribution of subjects. Following sample generation, we ask users to rate each image triplet from a scale of 1-to-5, with the following question: "Please rate the following image triplet from a scale of 1-to-5 (1 - unsatisfactory, 5 - very satisfactory) considering how realistic each image is and how naturally blended they are". The users are also supplied the foreground and background prompts used to generate the image triplet, for each method. We provide an example question from the conducted user study in Fig. 10.

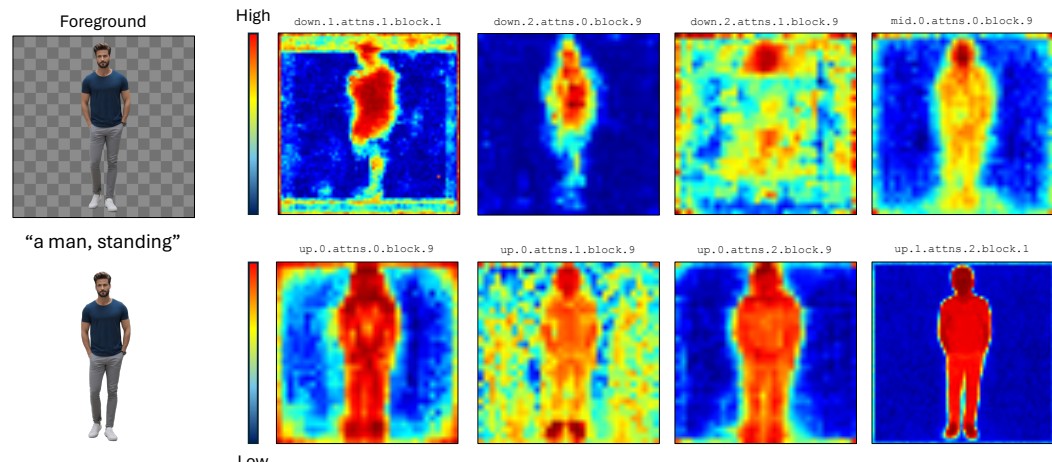

Figure 9: **Visualization of the structure priors from different self attention layers.** We visualize the structure priors extracted from different self attention layers of the foreground diffusion model, where the diffusion timestep is set as $t = 0.8T$. We visualize the structure priors from the self attention layer of each model block, follow the block definition of Frenkel et al. (2024). We follow the naming convention of diffusers (von Platen et al. (2022)). In all of our experiments, we use the structure prior from self attention layer `up.1.attns.2.block.1`.

---

**Algorithm 1** Extracting Structure Prior

---

**Require:** Foreground diffusion model $\epsilon_{\theta,FG}$, latent variable $z_t$, foreground conditioning $p_{FG}$

    **procedure** EXTRACTSTRUCTUREPRIOR($\epsilon_{\theta,FG}, z_t, p_{FG}$)

        # Retrieving the noise prediction(unused) and last self attention map

        $\epsilon_{pred}, m_{last} = \epsilon_{\theta,FG}(z_t, p_{FG})$

        $m = m_{last}$

        # Averaging over Attention Heads

        $m = \frac{\sum_{k=0}^{H} m_{k,i,j}}{H}$

        **for** $i \in m.shape(0)$ **do**

            # Assigning Sparsity Score

            $s_i = \frac{1}{\sum_{j=1}^{M} m_{i,j}^2}$

        **end for**

        # Converting Sparsity Score into Density Score

        s = 1 - NORMALIZE($s$)

        **return** s

    **end procedure**

---

**Algorithm 2** Extracting Content Confidence Prior

---

**Require:** Foreground diffusion model $\epsilon_{\theta,FG}$, hidden states $h$, foreground conditioning $p_{FG}$

    **procedure** EXTRACTCONTENTPRIOR($\epsilon_{\theta,FG}, h, p_{FG}$)

        # Retrieving Cross Attention Maps

        attn_out, attn_probs = $Attention_{\theta,FG}(h, p_{FG})$

        $n =$ attn_probs

        # Averaging over Attention Heads with `<EOS>` token

        $c = \frac{\sum_{k=0}^{H} n_{k,i,<EOS>}}{H}$

        **return** c

    **end procedure**

---

---

**Algorithm 3** Attention Blending

---

**Require:** Foreground diffusion model $\epsilon_{\theta,FG}$, RGB diffusion model $\epsilon_\theta$ foreground hidden states $h_{FG}$, blended hidden states $h_{Blended}$, background hidden states $h_{BG}$, foreground conditioning $p_{FG}$, background conditioning $p_{BG}$, boundary coefficient $d$, structure prior $s$

    **procedure** ATTNBLEND($\epsilon_{\theta,FG}, \epsilon_\theta, h_{FG}, h_{Blended}, h_{BG}, p_{FG}, p_{BG}, d, s$)

        # Layer Normalization for the cross attention layer

        $h_{norm,FG}, h_{norm,Blended}, h_{norm,BG} = $ LAYERNORMCROSSATTN($h_{FG}, h_{Blended}, h_{BG}$)

        $c = $ EXTRACTCONTENTPRIOR($\epsilon_{\theta,FG}, h_{norm,FG}, p_{FG}$)

        # Retrieving the Blending Masks

        $mask_{soft} = $ NORMALIZE($s * c$)

        $mask_{hard} = \sigma(d * (mask_{soft} - 0.5))$

        # Computing the Attention

        $a_{BG}, a_{Blended} = Attention_\theta([h_{BG}, h_{Blended}], p_{BG})$

        $a_{FG} = Attention_{\theta,FG}(h_{FG}, p_{FG})$

        # Blending Step

        $a'_{Blended} = a_{FG} * mask_{soft} + a_{Blended} * (1 - mask_{soft})$

        $a'_{FG} = a'_{Blended} * mask_{hard} + a_{FG} * (1 - mask_{hard})$

        **return** $a'_{FG}, a'_{Blended}, a_{BG}$

    **end procedure**

---

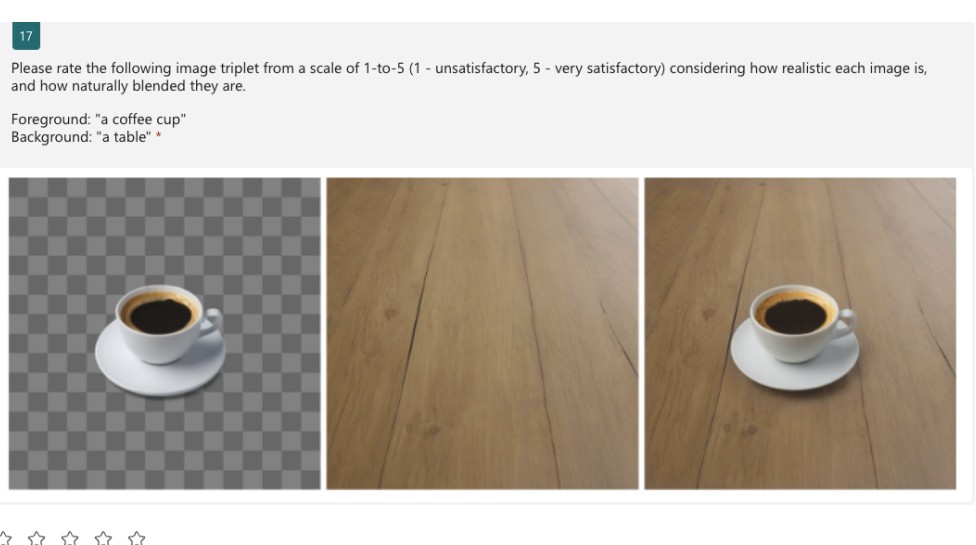

Figure 10: **Example Question from the User Study.** To evaluate the effectiveness our method perceptually, we conduct a user study over 40 generated image triplets. We provide an example question from this study for clarity. The users are shown an image triplet in the order of foreground, background and blended image and then asked to rate it from a scale of 1-to-5 (1 - unsatisfactory, 5 - very satisfactory).

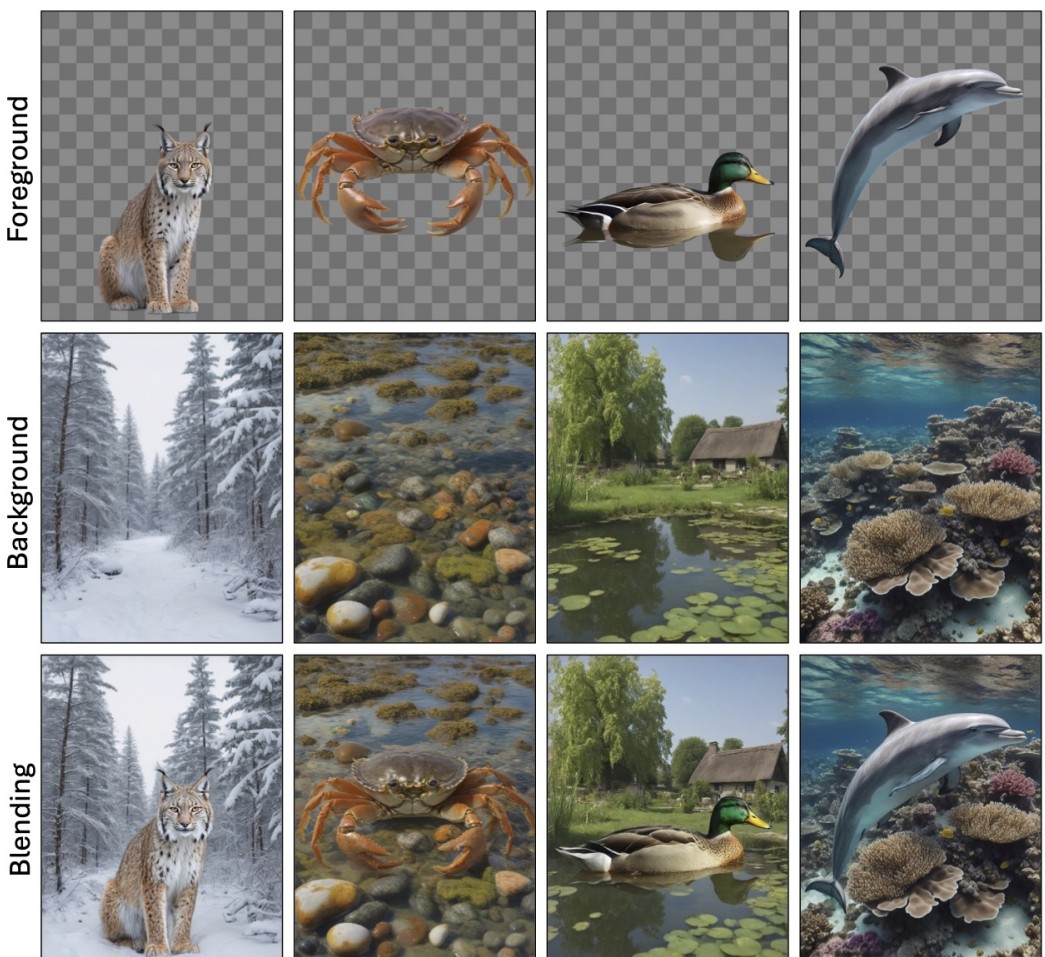

Figure 11: **Supplementary Generation Results with animal subjects.** Supplementary results with image resolution 896x1152. The foreground & background prompt pairs from left to right are: ("a lynx", "a snowy forest"), ("a crab", "a rocky tide pool"), ("a duck", "a village pond"), ("a dolphin", "a crystal-clear coral reef")

## A.6 SUPPLEMENTARY GENERATION RESULTS

In addition to the results provided in the main paper, we provide supplementary generation results in this section. Below, we include harmonized generations of a variety of subjects. We provide Fig. 11 to Fig. 20 as supplementary results.

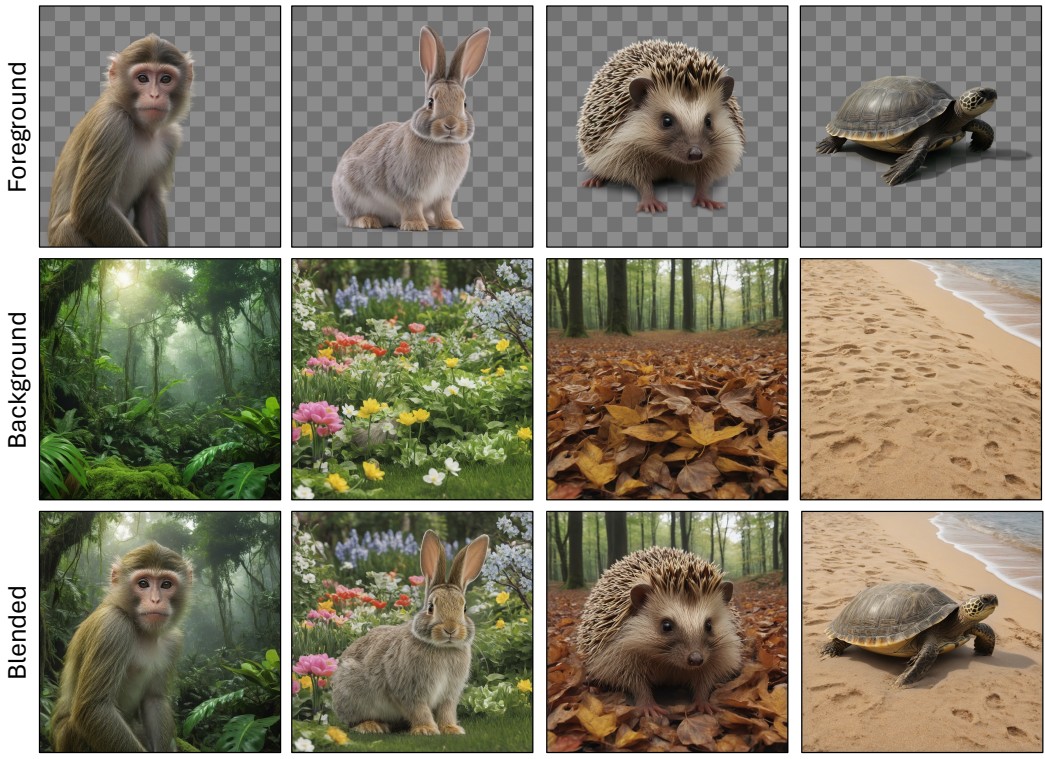

Figure 12: **Supplementary Generation Results with animal subjects.** Supplementary results with image resolution 1024x1024. The foreground & background prompt pairs from left to right are: ("a monkey", "a vibrant tropical rainforest"), ("a rabbit", "a backyard garden"), ("a hedgehog", "a forest floor covered in leaves"), ("a turtle", "a warm sandy beach")

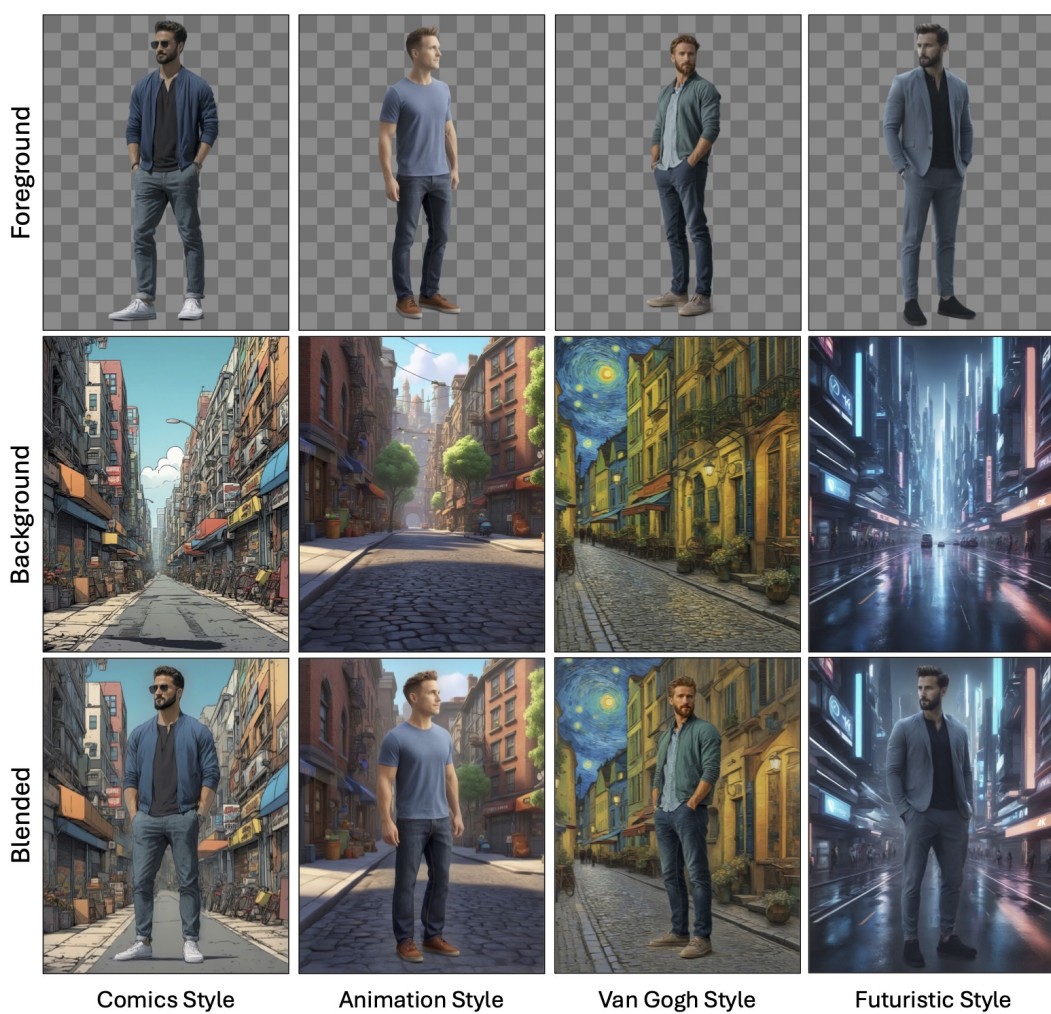

Figure 13: **Supplementary Generation Results with stylization prompts.** We provide additional examples with stylization prompts to demonstrate the harmonization capabilities of our method. For each image triplet, we generate the examples with the prompt set ("a man, standing", "a street, `style_name`") where `style_name` is "comics style" for the leftmost column. We provide the label (`style_name`) for each style under its respective image triplet. All images have the resolution of 896x1152.

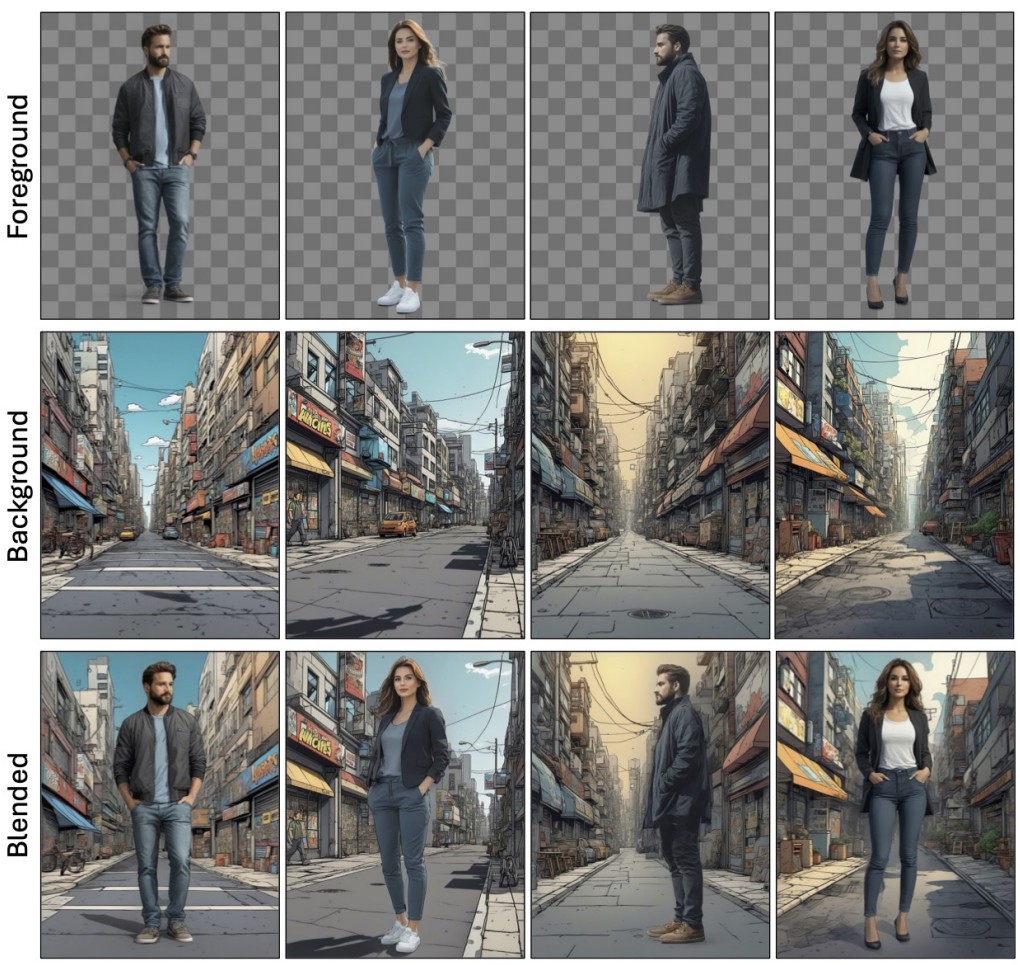

Figure 14: **Supplementary Generation Results for "comics" style.** To demonstrate the stylization capabilities of our layer harmonization approach, we provide additional results with the background prompt "a street, comics style". The resolution is 896x1152 for all of the images.

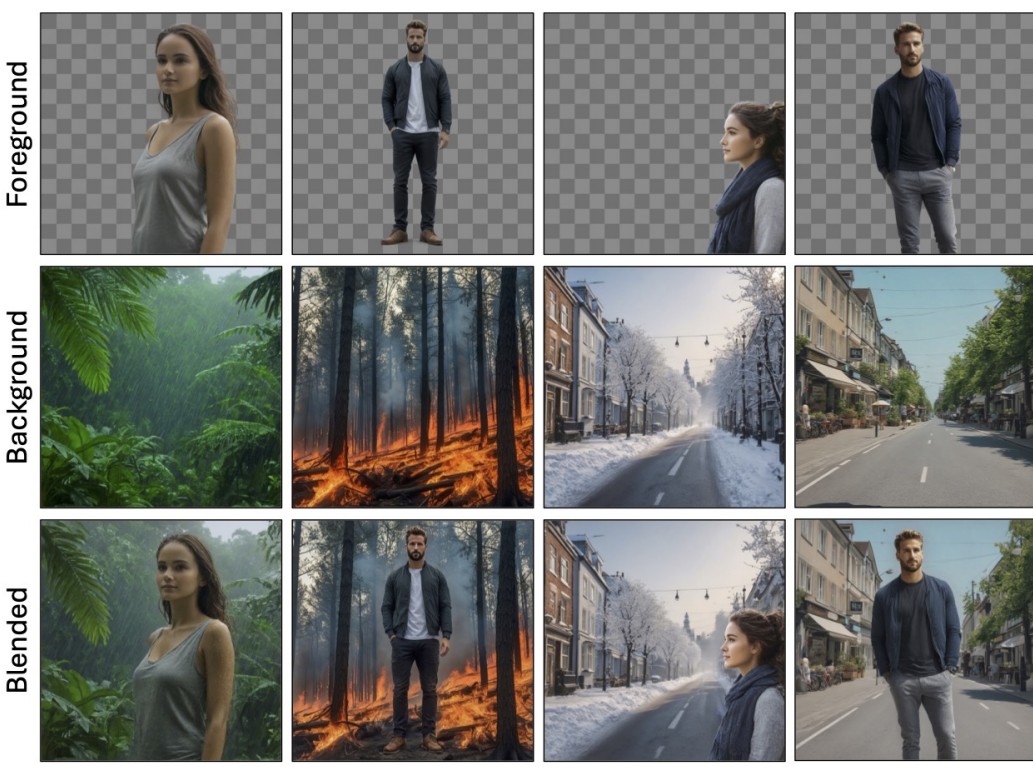

Figure 15: **Supplementary Generation Results with human subjects.** We provide additional examples with human subjects with different background prompts. The background prompts used are "a rainy jungle", "a forest in fire", "a street, winter time", "a street, daytime". Note that depending on the background prompt, the blending involves an interaction between the background and foreground (e.g. wetness in arm for the left-most image triplet). Image resolution is 1024x1024 for all examples.

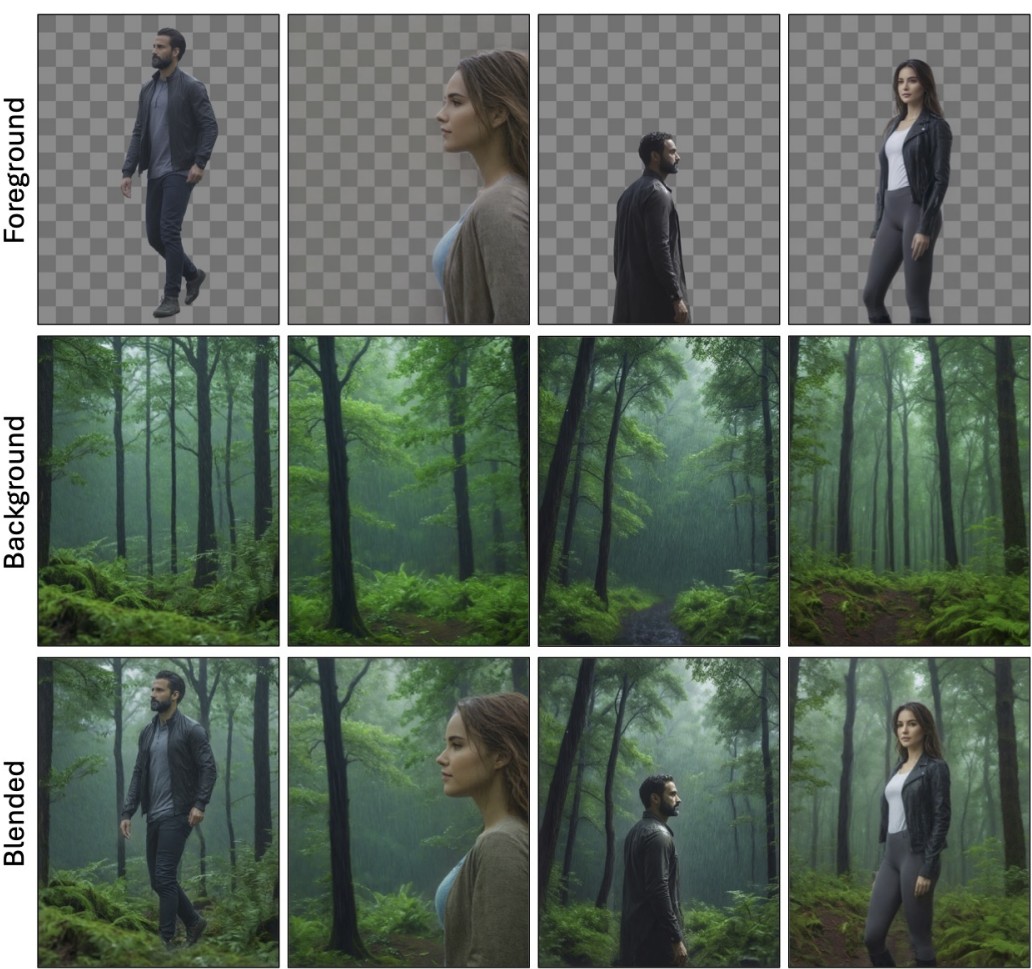

Figure 16: **Supplementary Generation Results for the background "a rainy forest".** For each of the images, the background prompt "a rainy forest" is used to generate the background image. As it can also be observed from the provided examples, the background creates an influence over the foreground (e.g. wetness effect on the human subjects). The image resolution is 896x1152 for all examples.

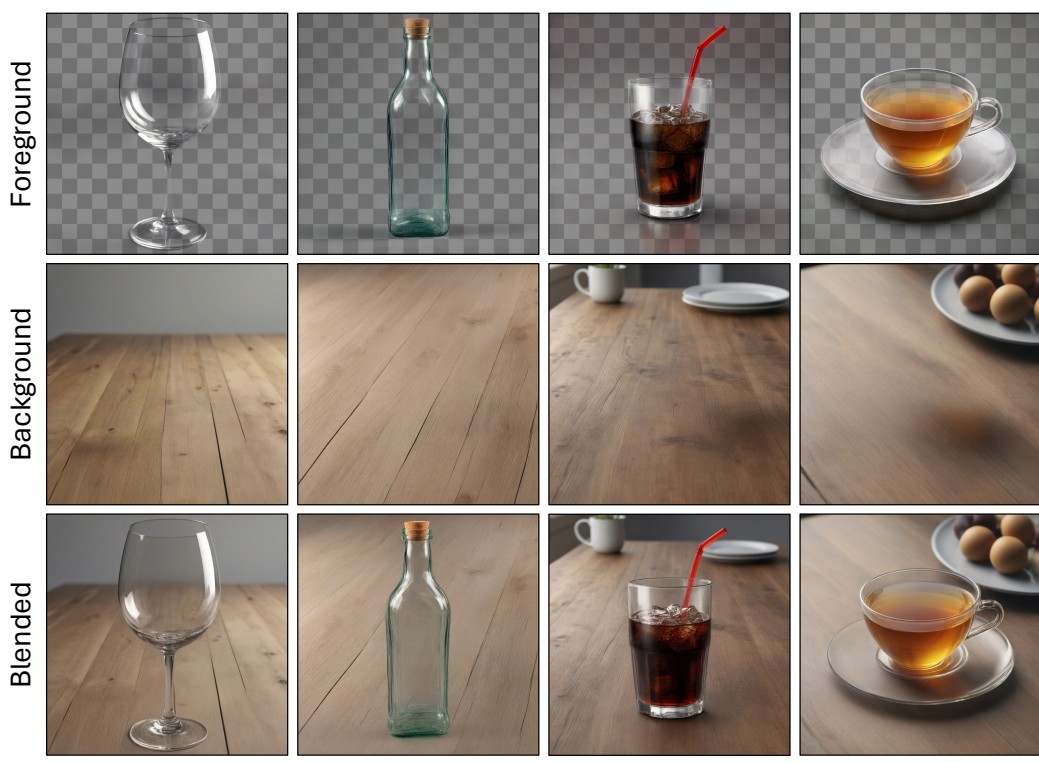

Figure 17: **Supplementary Generation Results for subjects with transparency property.** To demonstrate that our framework is able to preserve the transparency properties of layered image representations, we provide additional results here. With the background prompt "a table" we use the following foreground prompts: "a wine glass", "a glass bottle", "a cup filled with coke", "a cup of tea". All images have the resolution of 1024x1024.

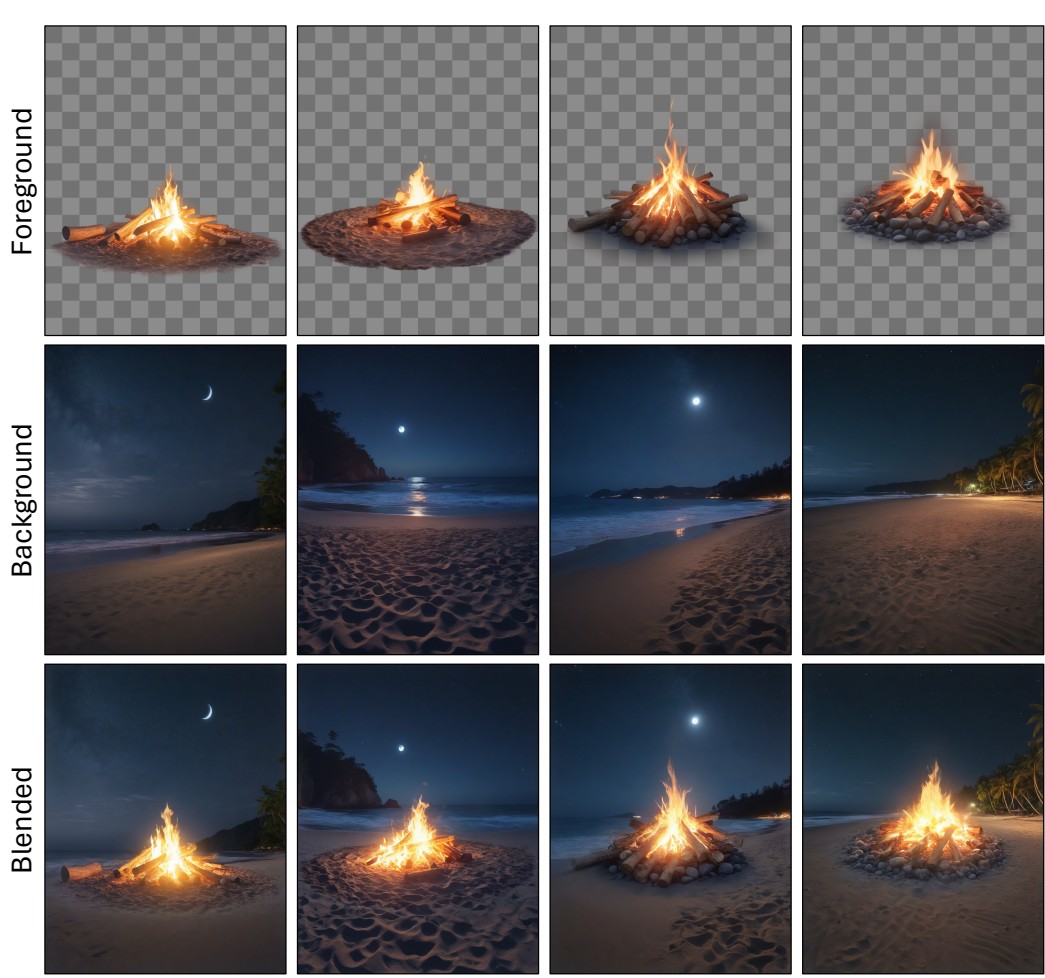

Figure 18: **Supplementary Generation Results for the subject "a campfire".** We provide additional generation results for the foreground prompt "a campfire" and background prompt "a beach, night time." The image resolution is 896x1152 for all examples.

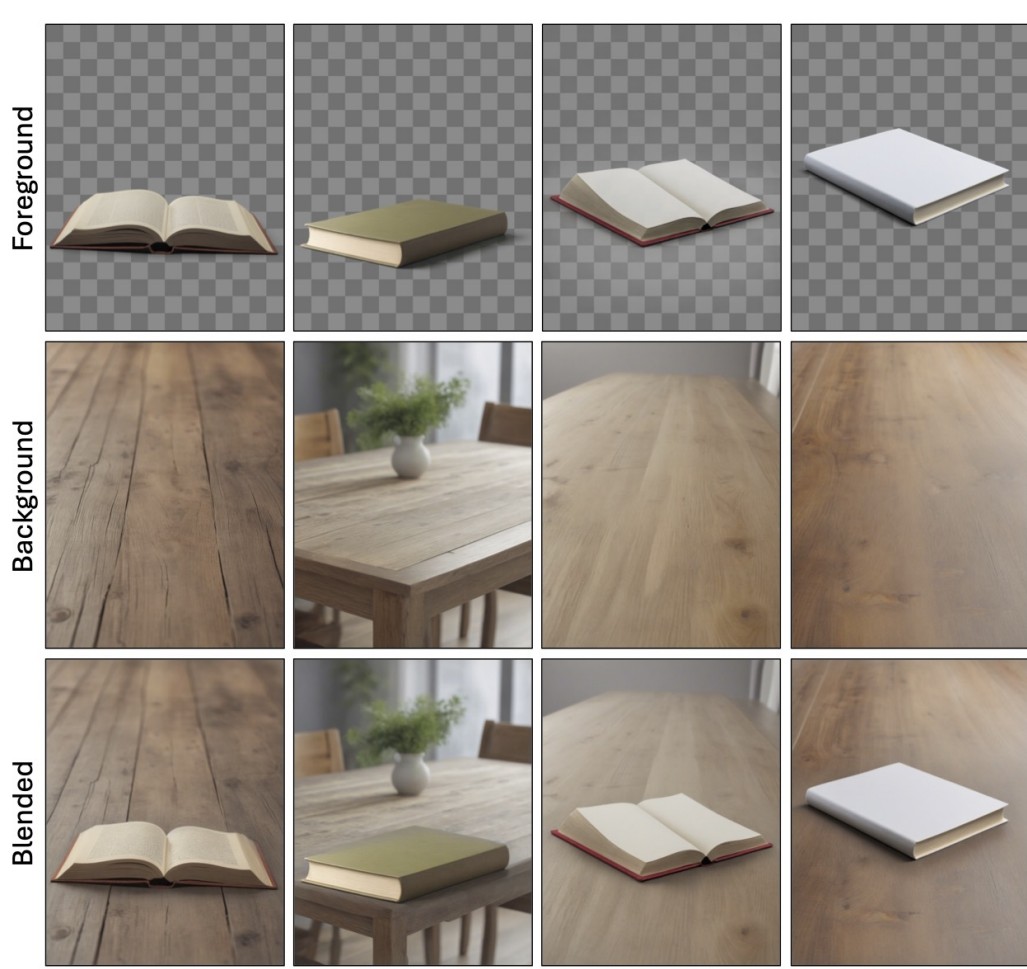

Figure 19: **Supplementary Generation Results for the subject "a book".** We provide additional generation results for the foreground prompt "a book" and background prompt "a table". The image resolution is 896x1152 for all examples.

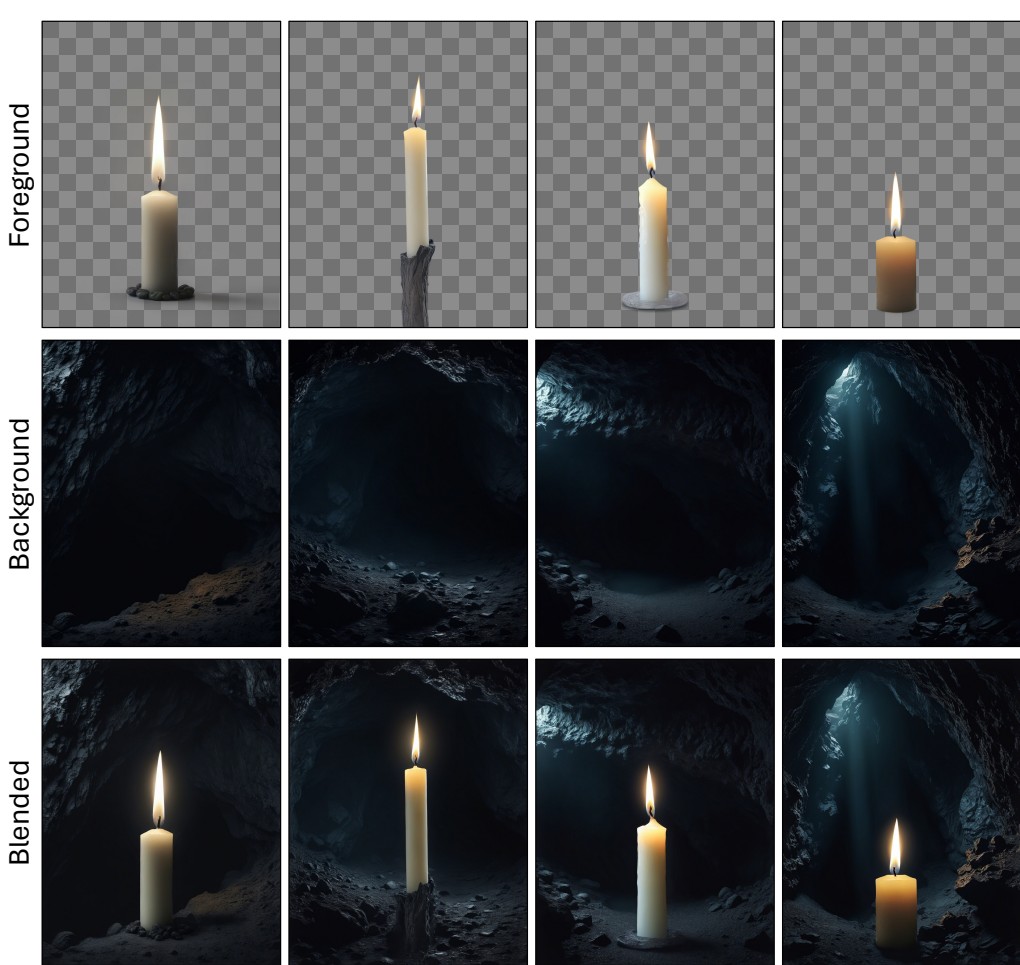

Figure 20: **Supplementary Generation Results for the subject "a candle".** We provide additional generation results for the foreground prompt "a candle" and background prompt "a dark cave". The image resolution is 896x1152 for all examples.

| Foreground | Background | Alpha Blending | Generative Blending |
|---|---|---|---|

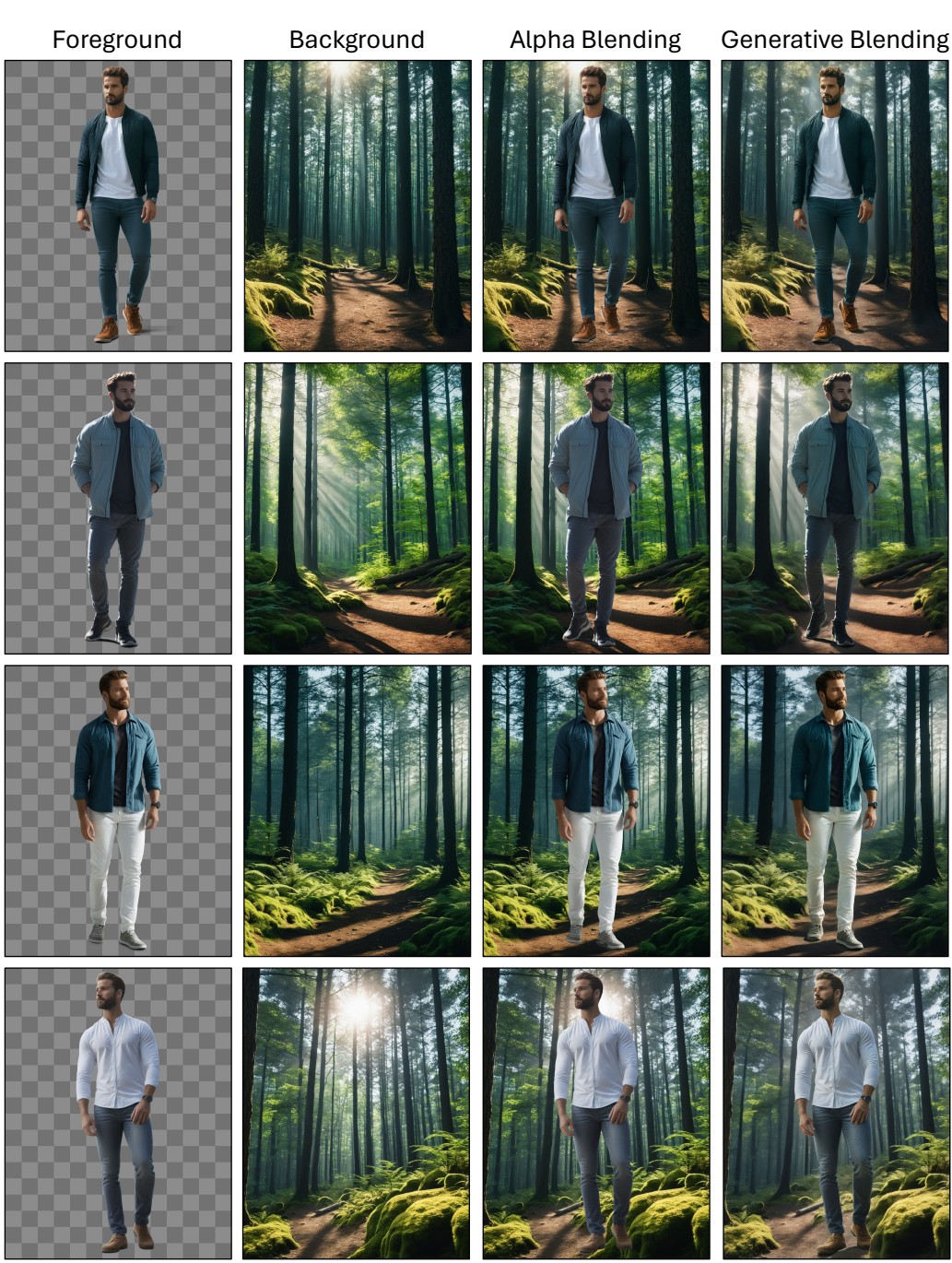

Figure 21: **Supplementary Generation Results for Grounding and Shadowing Effects.** We provide additional generation examples to demonstrate the grounding and shadowing capabilities of our framework. Our approach succeeds in both appropriate lighting compared to alpha blending (see rows 1, 2, 3), and can successfully ground the foreground on the background (row 4). We perform our generations with foreground prompt "a man, standing" and background prompt "a forest, daytime". The image resolution is 896x1152 for all examples.

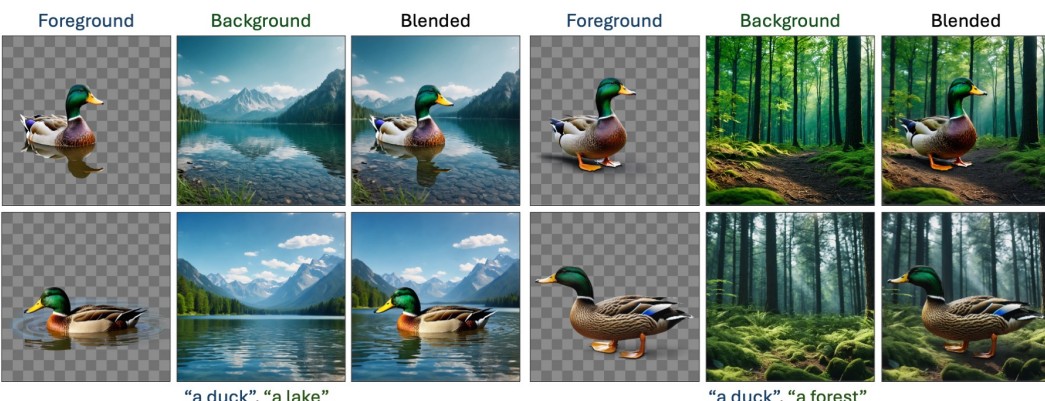

Figure 22: **Supplementary Generation Results Demonstrating Harmonization Capabilities.** We provide additional generation examples to demonstrate the harmonization capabilities of our approach. In each row, we provide triplets that are generated with the same initial seed, which the output resolution 1024x1024. As it can be observed from the provided examples, our framework can harmonize layers in a way that causes adaptations on the object shape, w.r.t. the background.

