# OpenReview forum: "LayerFusion: Harmonized Multi-Layer Text-to-Image Generation with Generative Priors"
_ICLR.cc/2025/Conference — Submitted to ICLR 2025_

### Official Review · Reviewer_878i · 2024-10-27

**Soundness:** 2
**Presentation:** 1
**Contribution:** 1
**Rating:** 3
**Confidence:** 3

**Summary:**

This paper introduces a training-free solution for blending the content of foreground and background layers to generate harmonious images. The image generation pipeline employs Latent Diffusion Models (LDMs) that adeptly produce images with two layers: a transparent foreground (RGBA) and a background (RGB). The authors introduce a novel attention-level blending scheme that effectively merges these layers using extracted masks.

**Strengths:**

1.The framework introduced in this paper represents a training-free approach for simultaneously generating layered content.

2.The paper introduces a novel attention-level blending scheme that uses extracted masks for seamless integration of foreground and background layers, resulting in cohesive and visually appealing compositions.

**Weaknesses:**

1.The writing requires polishing, as there are many errors in the article. For instance, in the section "Extracting Structure Prior" there are issues with the dimensions and the numerical domain of the attention probability map. Specifically, in Lines 195-196, the authors refer to the attention probability map with the notation $m \in \Re^{MxM}$. This appears to be a typographical error. It seems likely that the authors intended to write $m \in \mathbb{R}^{M \times M}$, which correctly denotes a matrix with dimensions M by M. Clarification and correction of such details are essential for maintaining the technical accuracy and clarity of the paper.

2.The evaluation presented in the paper appears to be insufficiently robust. The authors assert that their method enhances the "harmonization" of generated images, yet they rely on metrics such as the FID score, which primarily measures image quality, and the CLIP score, which assesses text-image alignment. Unfortunately, neither of these metrics directly evaluates the "harmonization" of images. Based on my understanding, the authors could significantly strengthen their evaluation by leveraging existing image segmentation models to perform segmentation tasks on the generated images. By assessing the accuracy of these downstream tasks, they would be able to more effectively evaluate the quality of image harmonization, providing a more direct measure of how well the foreground integrates with the background in the generated images.

3.Benchmark is not clear; the benchmark dataset utilized for the evaluation is not explicitly specified. Could the authors please clarify which benchmark dataset was used?

4.I think the problem addressed in the paper is not suitable for this conference. The method proposed by the authors employs a manually designed blending technique to improve the integration of foreground and background elements in image blending. As such, it is more suitable for conferences in the multimedia domain.

**Questions:**

1.What is the definition of "harmonization" of generated image and why is important？

---

> ### Author Response · Authors · 2024-11-23
>
> We thank the reviewers for their constructive criticism and acknowledging the novelty in the attention-blending scheme introduced. Please find our responses to the mentioned weaknesses and questions below:
>
> **W1: Correction on Notation Used**
>
> We appreciate the reviewer pointing out the typographical error in the notation for the attention probability map. The correct notation should indeed be $m \in \mathbb{R}^{M \times M}$, representing a matrix with dimensions M×MM \times MM×M. We correct this error in the uploaded rebuttal.
>
> **W2: Evaluation**
>
> In our quantitative evaluation (Section 4.2), we focus on preserving the distributions of the foreground and background generation models. To complement this, we conduct a perceptual study to evaluate the harmonization quality of the blended images.
> Regarding the suggestion to use image segmentation models for evaluation, we acknowledge the potential of segmentation-based methods in assessing foreground-background integration. However, current segmentation models face challenges in accurately extracting RGBA images with their corresponding alpha channels, particularly for generated content. We discuss these limitations in Figure 6, highlighting that such methods may experience difficulties in extracting RGBA content.
>
> **W3: Evaluation Benchmark**
>
> Since our task involves generating RGBA foregrounds, RGB backgrounds, and their harmonious blending, our evaluation is conducted on generated image triplets. Section 4.2 (Foreground & Background Quality - claims highlighted in yellow in the uploaded rebuttal) specifies that these triplets are derived from our foreground and background generation models.
>
> **W4: Problem Addressed in the Paper**
>
> While our method addresses challenges in image blending, it also provides valuable insights into the internal representations learned by diffusion models. Specifically, we introduce:
> - A novel way of interpreting self-attention maps through a sparsity score for structure extraction.
> - The first demonstration of attention-level blending for visual tasks such as image blending.
>
> These contributions extend beyond traditional blending tasks by offering new perspectives on the interpretability and utilization of diffusion models. We believe this aligns with ICLR’s focus on advancing machine learning methods and their applications.
> Furthermore, the ICLR community has a history of recognizing work on vision tasks that explore creative generation and interpretability, as evidenced by prior publications ([3], [4], [5], [6]).
>
> [3] IMPUS: Image Morphing with Perceptually Uniform Sampling Using DIffusion Models, https://openreview.net/pdf?id=gG38EBe2S8, ICLR 2024
>
> [4] DiffEdit: Diffusion-Based Semantic Image Editing with Mask Guidance, https://openreview.net/pdf?id=3lge0p5o-M-, ICLR 2023
>
> [5] Diffusion in Diffusion: Cyclic One-Way Diffusion For Text-Vision-Conditioned Generation, https://openreview.net/pdf?id=ePOjNlOjLC, ICLR 2024
>
> [6] Generating Images with 3D Annotations Using Diffusion Models, https://openreview.net/pdf?id=XlkN11Xj6J, ICLR 2024
>
> **Q1: Definition of "Harmonization" and Its Importance**
>
> Harmonization, in the context of our work, refers to the seamless integration of the foreground and background layers such that the final blended image appears cohesive and visually natural. This is particularly important for applications in creative workflows, where spatial editing, object movement, and compositional adjustments rely on well-integrated layers. To clarify the effectiveness of harmonized blending, we provide supplementary results in Fig. 21, that highlights various aspects of harmonization such as shadowing, lighting and grounding.

---

> ### Author Response · Authors · 2024-11-25
>
> Dear Reviewer 878i,
>
> Since the deadline for author discussions are approaching, we kindly ask if you may provide any feedback or additional questions you have. We will do our best to address these issues before the rebuttal period closes.
>
> Thank you,
>
> Paper 1337 authors

---

> > ### Author Response · Authors · 2024-12-01
> >
> > Dear Reviewer,
> >
> > Just a friendly reminder that tomorrow is the deadline for reviewers to submit their comments. If you have any outstanding questions or need further clarification, we are more than happy to provide additional details.
> >
> > Should our responses have addressed your concerns satisfactorily, we kindly ask that you consider raising the score.
> >
> > Thanks again for all your constructive comments.
> >
> > Thanks,
> > Authors of Paper 1337

---

> ### Author Response · Authors · 2024-12-03
>
> Dear Reviewer 878i,
>
> Thank you for your contributions to the reviewing process until today. As a final reminder, please consider revising your final score given our responses and the revised manuscript regarding the concerns stated by your review.
>
> We hope to receive your response, as we are now approaching to the end of the review period.
>
> Thanks,
>
> Paper 1337 authors

---

### Official Review · Reviewer_MWZY · 2024-11-01

**Soundness:** 2
**Presentation:** 3
**Contribution:** 3
**Rating:** 5
**Confidence:** 4

**Summary:**

This paper introduces a novel image generation pipeline based on Latent Diffusion Models that focuses on generating layered content, specifically images with a foreground layer (RGBA) and a background layer (RGB). Unlike previous methods that generate these layers sequentially, the proposed approach introduces a harmonized generation mechanism that allows for dynamic interactions between the layers, resulting in more coherent outputs. The paper also presents a novel attention-level blending scheme that utilizes extracted masks to seamlessly blend the foreground and background layers, ensuring cohesive interaction and aesthetically pleasing compositions. Through extensive qualitative and quantitative experiments, the paper demonstrates the effectiveness of the proposed method in generating high-quality, harmonized layered images, outperforming baseline methods in terms of visual coherence, image quality, and layer consistency across various evaluation metrics.

**Strengths:**

The key advantages of this paper can be summarized as:

1. Innovative Layered Image Generation: The paper introduces a novel pipeline based on Latent Diffusion Models (LDMs) that generates layered content, including RGBA foreground and RGB background, addressing a gap in single-layer image generation.
2. Harmonized Generation Mechanism: The proposed approach enables dynamic interactions between layers, resulting in coherent and visually appealing outputs crucial for graphic design, animation, and digital art.
3. Attention-Level Blending Scheme: A unique blending scheme uses masks to seamlessly blend layers, ensuring cohesive interaction and natural compositions.
4. Extensive Experimental Validation: Qualitative and quantitative experiments show significant improvements in visual coherence, image quality, and layer consistency compared to baseline methods.

**Weaknesses:**

The Harmonized Generation Mechanism seemingly focuses more on the attention given to the foreground structure, but in terms of content understanding, there appear to be some issues based on the results shown in Figure 4. For instance, in the third column, the car and the tire tracks on the street look out of place. In the fourth and sixth columns, shadows are not properly reconstructed. The fifth column gives a more "sticker-like" appearance. The authors can show more additional experiments and analyses what could help identify the root cause of these problems, such as an ablation study on different components of the harmonization mechanism.

**Questions:**

1. Which Parameters in SD need Fine-Tuning? Have you ever investigated which layers or attention mechanisms in the SD model are most crucial for improving the layered generation process.
2. How the  approach in this paper to harmonization compares to PCT-Net's tone consistency techniques？Have you ever considered incorporating similar mechanisms?

---

> ### Author Response · Authors · 2024-11-23
>
> We thank the reviewers for acknowledging the novelty introduced and constructive criticism. Please find our responses corresponding to the concerns raised:
>
> **W1: Dependency on Foreground Structure**
>
> We appreciate the reviewer highlighting the limitations in harmonization quality in certain examples (e.g., Figure 4). As noted, our method relies on a baseline model for RGBA image generation. This baseline tends to favor generating large, centered objects, which can limit its ability to produce fully harmonized outputs in all scenarios.
>
> Despite this, our method demonstrates improvements in generating backgrounds that physically align with the foreground in terms of spatial positioning and contextual coherence. For example, the snow effect in Figure 4 showcases meaningful interactions between the foreground and background. We acknowledge that using a baseline model introduces some limitations, which we discuss in section A.2 included in the supplementary material. For ease of reference, we highlight the mentioned discussion in yellow in the rebuttal uploaded.
>
> In addition, to better demonstrate the harmonization capabilities of our approach, we provide additional results involving shadows, lighting and grounding in Fig. 21 in the revised manuscript (highlighted in yellow). Here, we additionally provide comparisons with alpha blending, which highlights the advantages of our attention-level blending approach.
>
> **Q1: Trained Parameters**
>
> Our proposed method is training-free with respect to both the RGBA generation model (LayerDiffuse) and the RGB generation model (SDXL). However, the RGBA baseline model does require fine-tuning of the attention layer weights using a LoRA-based approach to adapt it for foreground generation. Note that, we use the pre-trained weights of [2], and do not perform any additional training.
>
> **Q2: Comparison to Methods Like PCT-Net**
>
> Thank you for suggesting a potential extension of our work. Our method focuses on a training-free generation pipeline, adapting it to methods like PCT-Net, which involve training for consistent tone generation across layers is beyond the scope of this work.
>
> [2] Transparent Image Layer Diffusion using Latent Transparency, https://arxiv.org/abs/2402.17113, SIGGRAPH 2024

---

> ### Author Response · Authors · 2024-11-25
>
> Dear Reviewer MWZY,
>
> Since the deadline for author discussions are approaching, we kindly ask if you may provide any feedback or additional questions you have. We will do our best to address these issues before the rebuttal period closes.
>
> Thank you,
>
> Paper 1337 authors

---

> > ### Author Response · Authors · 2024-12-01
> >
> > Dear Reviewer,
> >
> > Just a friendly reminder that tomorrow is the deadline for reviewers to submit their comments. If you have any outstanding questions or need further clarification, we are more than happy to provide additional details.
> >
> > Should our responses have addressed your concerns satisfactorily, we kindly ask that you consider raising the score.
> >
> > Thanks again for all your constructive comments.
> >
> > Thanks,
> > Authors of Paper 1337

---

> ### Author Response · Authors · 2024-12-03
>
> Dear Reviewer MWZY,
>
> Thank you for your contributions to the reviewing process until today. We would like to make a final reminder on the possibility of revising your score, given the concerns and questions addressed in the comments we provide and in the revised manuscript.
>
> We hope to receive your revision on your review, as we are now approaching to the end of the review period.
>
> Thanks,
>
> Paper 1337 Authors

---

### Official Review · Reviewer_jFaU · 2024-11-02

**Soundness:** 2
**Presentation:** 2
**Contribution:** 2
**Rating:** 5
**Confidence:** 4

**Summary:**

This paper proposes a novel image generation pipeline based on Latent Diffusion Models (LDMs) that generates images with two layers: a foreground layer (RGBA) with transparency information and a background layer (RGB).

**Strengths:**

This paper proposes a novel image generation pipeline based on Latent Diffusion Models (LDMs) that generates images with two layers: a foreground layer (RGBA) with transparency information and a background layer (RGB).

**Weaknesses:**

I find the task setting somewhat confusing in terms of its purpose. If a foreground image is generated and then a blended image is produced, what additional flexibility does this provide compared to generating a complete image directly? Since both the foreground and background images are generated rather than provided by the user, unlike other blending tasks, the setup seems unusual and lacks clear practical applications or value.

The method’s contributions seem limited. As the model is training-free, operations such as extracting attention maps, generating masks, and performing attention blending are already widely used as general techniques in the AIGC field, and therefore cannot be considered significant contributions.

As this is a niche task defined by the authors, it feels quite specific, making it challenging to evaluate the effectiveness of the experiments and to perform meaningful comparisons.

**Questions:**

see weakness

---

> ### Author Response · Authors · 2024-11-23
>
> We thank the reviewers for acknowledging the novelty introduced in our method. Please find our responses on the raised concerns below:
>
> **W1: Comparison with Generation Using Composite Prompts**
>
> Thank you for raising this point. While it is indeed possible to generate composite images using a single prompt, our method offers significant advantages by also generating an alpha channel alongside the foreground image. This provides users with flexibility in spatial editing tasks such as moving or resizing objects, which would otherwise be difficult with a directly generated composite image.
>
> Furthermore, the separation of foreground and background layers makes our approach a potential data generation method for tasks such as image inpainting and object removal, where having background and foreground layers are highly beneficial.
>
> **W2: Contributions**
>
> We understand the concern about the perceived novelty of using attention maps for blending, as this has been explored in prior work. However, our method introduces two key innovations:
> - **Structure Prior from Self-Attention Maps**: We propose a novel interpretation of self-attention maps as structure priors, which is detailed in Section 3.2 (highlighted in yellow) and Algorithm 1. This provides additional guidance during blending, improving spatial coherence.
> - **Attention-Level Blending Mechanism**: To the best of our knowledge, our method is the first to perform blending at the attention level. Unlike methods such as Blended Latent Diffusion, which rely on latent variables for blending, our approach utilizes attention outputs, enabling finer control over the blending process.
>
> These contributions expand the potential of attention mechanisms in generative tasks, which we address in our key contributions listed at the introduction section. For the ease of visibility, we highlight these contributions in yellow, in the introduction section.
>
> **W3: Evaluation**
>
> As the task of image triplet generation with RGBA foreground and RGB background is relatively new, our method serves as a first approach to enable inference-time blending for such images. While this limits direct comparisons, we have included:
> - Results from other RGBA generation-based approaches (Table 1).
> - Comparisons with harmonization methods to demonstrate the effectiveness of our blending strategy (Figure 7 - highlighted in yellow).
> In addition, we conducted a user study to address the limitations of quantitative metrics in evaluating perceptual quality. The results of this perceptual study indicate that our method outperforms the baseline approach requiring additional training for harmonization tasks.

---

> ### Author Response · Authors · 2024-11-25
>
> Dear Reviewer jFaU,
>
> Since the deadline for author discussions are approaching, we kindly ask if you may provide any feedback or additional questions you have. We will do our best to address these issues before the rebuttal period closes.
>
> Thank you,
>
> Paper 1337 authors

---

> ### Comment · Reviewer_jFaU · 2024-11-26
>
> Thanks to the author's reply, although it is said that on this task, this method is the first to perform blending at the attention level, in other AIGC fields, similar operations such as attention level replacement, panning, deflation, weighted fusion, etc., I encountered many times at the review of manuscripts from 2023 to now.
>
> To be honest, the quality of this work is good, but still not good enough for a field like AIGC. I suggest authors try simpler conferences or journals. Or try to explore the value of applying this work in other fields, such as applying it to traditional high-level, low-level tasks or dataset construction, etc., to show its value in serving other tasks, and then submit it to the top conferences.

---

> ### Author Response · Authors · 2024-11-28
>
> Thanks for your suggestions regarding possible improvements on our work. While acknowledging the use of attention features on other applications in AIGC field, we refer to the operation of performing harmonization on attention level as our novelty, instead of utilization of attention features overall.
>
> In addition, our method serves as the first approach to utilize the attention outputs as a tentative alpha channel estimate. Note that the baseline model [1] does not provide any alpha channel representation during the generation process, but outputs the alpha channel from the final foreground image. As our harmonization approach aims to output a foreground and background that are compatible with each other, such an alpha channel prediction is insufficient where intermediate blending steps are necessary for interactions between layers. To resolve this, we propose using generative priors from the attention layers to get a coarse alpha channel prediction in the form of a mask (please refer to Sec. 3.3 and Alg. 3), to be able to perform the blending. This use case of the attention maps also serves as a novelty of our method, compared to other applications in the field of AIGC.
>
> Regarding the novelty introduced with the attention scheme we use for blending, we will clarify the introduced novelty with attention blending in the final version of our paper.
>
> [1] Transparent Image Layer Diffusion using Latent Transparency, https://arxiv.org/abs/2402.17113, SIGGRAPH 2024

---

> > ### Author Response · Authors · 2024-12-01
> >
> > Dear Reviewer,
> >
> > Just a friendly reminder that tomorrow is the deadline for reviewers to submit their comments. If you have any outstanding questions or need further clarification, we are more than happy to provide additional details.
> >
> > Should our responses have addressed your concerns satisfactorily, we kindly ask that you consider raising the score.
> >
> > Thanks again for all your constructive comments.
> >
> > Thanks,
> > Authors of Paper 1337

---

### Official Review · Reviewer_jYPb · 2024-11-02

**Soundness:** 3
**Presentation:** 3
**Contribution:** 3
**Rating:** 6
**Confidence:** 3

**Summary:**

Layered content generation is crucial for creative workflows in various image fields. This paper proposes a layered content design method based on Latent Diffusion Models (LDMs) and employs a harmonized attention mechanism to enhance image generation quality.

**Strengths:**

1. Proposed a blending generation mechanism that enables dynamic interactions between different layers in layered content generation.

2. Utilized attention mechanisms to allow adjustments between different layer images, enhancing the realism of the generated images.

**Weaknesses:**

1. The foreground generation model relies on a pretrained model from existing work, which may conflict with the current mechanism.

2. There is a lack of quantitative metrics for evaluating the blending images.

**Questions:**

It is hoped that there will be quantitative metrics for evaluating the blending images.

---

> ### Author Response · Authors · 2024-11-23
>
> We thank the reviewers for acknowledging the contributions of our work and your constructive feedback. Please find our responses to the mentioned concerns below:
>
> **W1: Using a Pretrained Model**
>
> Thank you for pointing this out. In our work, we focus on leveraging the capabilities of existing pretrained models to extract meaningful information for the blending task. This approach eliminates the need for collecting RGBA foreground data paired with RGB backgrounds, which is an expensive and resource-intensive process. By relying on pretrained models, we can bypass these challenges while demonstrating the flexibility and potential of our framework. Please see section A.2 in the supplementary material for discussions regarding the limitations caused by the pretrained model. We highlight this discussion with the color yellow, for the ease of visibility in the rebuttal uploaded.
>
> **W2: Lack of Quantitative Metrics and Evaluation**
>
> We acknowledge the importance of quantitative metrics in evaluating the blending quality. Since the task of image generation involving RGBA foregrounds and RGB backgrounds is relatively new, there is limited prior work for direct comparison. However, we have included comparisons with methods capable of performing this task, as well as harmonization models, which address overlapping but distinct challenges. In addition, to address the gap regarding the lack of quantitative metrics available for the task, we also conduct a user study as a perceptual evaluation, which shows the effectiveness of our blending approach.

---

> > ### Comment · Reviewer_jYPb · 2024-11-27
> >
> > Thanks for the author's response. After reading all the reviewers' comments and the author's corresponding replies, I believe this paper is good, but there is still room for improvement. I will maintain my score and hope that the author continues to make progress and further refine this work.

---

> ### Author Response · Authors · 2024-11-25
>
> Dear Reviewer jYPb,
>
> Since the deadline for author discussions are approaching, we kindly ask if you may provide any feedback or additional questions you have. We will do our best to address these issues before the rebuttal period closes.
>
> Thank you,
>
> Paper 1337 authors

---

### Official Review · Reviewer_FJ9h · 2024-11-04

**Soundness:** 2
**Presentation:** 3
**Contribution:** 2
**Rating:** 5
**Confidence:** 4

**Summary:**

The paper introduces a novel image generation pipeline, which splits the generation of the whole image into foreground in RGBA format and background layers in RGB format, providing flexible editing and composition ability. Specifically, this paper proposes a harmonized generation mechanism framework that enables dynamic interactions between the foreground and background layers. The framework first extracts attention mask as the structure prior from the self-attention, and then extracts the content confidence map from the cross-attention in the foreground model. In the blending pipeline, the blended image and the foreground image are updated respectively based on the soft and hard mask. Overall, the proposed method result in more coherent outputs and more flexible editing pipelines compared to traditional sequential generation methods.

**Strengths:**

1. Splitting the image generation into foreground and background settings is more aligned with real-world application scenarios, facilitating easier editing for users.

2. The harmonized mechanism based on attention masks can effectively handle interactions between foreground and background, such as lighting and style.

3. The paper is well-written and clearly articulated.

**Weaknesses:**

1. Unstable interaction. In the Fig.8(b), the "glass" and "woman" sample repectively lack shadow and sufficiently strong Van Gogh style in the blending image. Does the threshold of the mask boundary affects the results? Please provide more details on how the mask boundary threshold is determined, and discuss the impact of boundary threshold on the blending quality across different types of objects and styles.

2. Multi-characters. The results in the paper are only contained one character. Do you test your method on prompts with multiple foreground objects? Please provide examples and discuss any challenges or limitations they encountered.

3. Further interaction. How does the pipeline perform on the physical interaction, such as "a man holding a glass of wine" or "a kid sitting on the chair"? Please provide examples of the performance on prompts involving physical interactions between foreground and background elements, and iscuss any specific challenges or adaptations needed for such cases.

**Questions:**

Please see the weakness.

---

> ### Author Response · Authors · 2024-11-23
>
> Thank you for your insightful feedback, please find the answers related to the adresses concerns below:
>
> **W1: Unstable Interaction**
>
> We appreciate the reviewer highlighting the effect of the mask threshold. To clarify, the boundary coefficient is used primarily to ensure object completeness rather than to control stylistic or lighting changes. The main effect of the mask boundary coefficient (denoted as d in the calculations) is to ensure complete coverage of the foreground during blending, as demonstrated in the ablations provided in Figure 8(d) (explanations highlighted with yellow in the uploaded rebuttal).
>
> Rather than applying a constant threshold, we propose an adaptive masking strategy based on the distribution of changing mask values across iterations, allowing for a soft decision boundary. This method ensures flexibility and robustness during masking. Regarding the boundary coefficient, we determine the initial boundary coefficient value by integrating the masking strategy proposed by [1]. The algorithm for mask construction is detailed in Section 3.3 and elaborated further in Algorithm 3 of the supplementary material.
>
>
> **W2: Multi-Characters**
>
> Thank you for pointing out the importance of multi-character generation. As the baseline model used for foreground generation is biased towards generating a single, centered object, our work has focused on single-foreground scenarios to evaluate the effectiveness of our approach. However, we note that the generated foreground is in RGBA format, which allows for manual construction of scenes involving multiple objects.
>
> We address such limitations in the supplementary material, precisely in section A.2.
>
> **W3: Further Interaction**
>
> We acknowledge that scenes involving physical interactions and occlusions are a limitation of our current approach. Our method relies on attention outputs to extract a tentative alpha channel estimate, which primarily provides positional information about the generated object. Overcoming this limitation would require baseline models capable of explicitly modeling physical interactions, as our LayerDiffuse framework focuses on generating objects in isolation.
> We discuss such limitations that occur due to the baseline model used in section A.2 of the supplementary material.
>
> For ease of reference, we highlight the discussions provided in section A.2 in yellow.
>
> [1] Diffusion Self-Guidance for Controllable Image Generation, https://arxiv.org/abs/2306.00986, NeurIPS 2023

---

> ### Author Response · Authors · 2024-11-25
>
> Dear Reviewer FJ9h,
>
> Since the deadline for author discussions are approaching, we kindly ask if you may provide any feedback or additional questions you have. We will do our best to address these issues before the rebuttal period closes.
>
> Thank you,
>
> Paper 1337 authors

---

> > ### Comment · Reviewer_FJ9h · 2024-11-27
> >
> > Thank you for your response, which has addressed most of my concerns. The paper is well-motivated since generating foreground and background separately can enhance controllability. However, compared to models that generate foreground and background simultaneously, there is still a significant gap in terms of physical interaction and sufficient style consistency between the foreground and background. These issues remains to be addressed by further methods. Besides, most of the foreground shadows appears lack sufficient harmonization with the background. For example, what does the shadow of the duck in Fig.1 look like if the duck is put on the ground?

---

> > > ### Author Response · Authors · 2024-11-27
> > >
> > > Thank you for your comments and suggestions regarding future directions of our work. Regarding the harmonization capabilities, that are addressed over the duck subject in Fig. 1, our method is able to adapt the foreground subject into the background scene. To demonstrate this capability of our framework further, we provide additional examples with the "duck" subject in Fig. 22, where the foreground is placed over scenes in both swimming and standing setups.

---

> > > > ### Comment · Reviewer_FJ9h · 2024-11-28
> > > >
> > > > Thanks for the additional cases. What I am particularly curious about is whether the proposed method is robust enough to eliminate unnatural shadows. For instance, in Fig.22, the duck in the top left has a water reflection. If a desert background is provided, would the unnatural water reflection still exist?

---

> > > > > ### Author Response · Authors · 2024-11-28
> > > > >
> > > > > Thank you for raising this important issue regarding the harmonization performance. In the examples that we provided in Fig. 22, we use the same initial latents to generate the examples. As it can be seen over the provided supplementary cases, the background condition results in adaptation in the foreground shape (despite using the same initial latent and same generation condition), where the reflection effect is only visible in the lake setting.

---

> > > > > > ### Author Response · Authors · 2024-12-01
> > > > > >
> > > > > > Dear Reviewer,
> > > > > >
> > > > > > Just a friendly reminder that tomorrow is the deadline for reviewers to submit their comments. If you have any outstanding questions or need further clarification, we are more than happy to provide additional details.
> > > > > >
> > > > > > Should our responses have addressed your concerns satisfactorily, we kindly ask that you consider raising the score.
> > > > > >
> > > > > > Thanks again for all your constructive comments.
> > > > > >
> > > > > > Thanks,
> > > > > > Authors of Paper 1337

---

### Author Response · Authors · 2024-11-23

**Dear Reviewers and ACs,**

Thank you for your time and attention in reviewing our paper! The constructive criticism that we received through your reviews have been invaluable in refining our work where we address the raised concerns as comments.

Taking the suggestions and criticism made during the review process, we updated the previous version of our manuscript with necessary clarifications and supplementary examples. We acknowledge that the majority of the concerns regarding our paper address the metrics used for evaluation, qualitative results presented in our paper, and the novelty introduced in our method.

We address the following concerns in both the rebuttal comments and the revised manuscript, where the sections mentioned in the rebuttal are highlighted with yellow in the revised version.
1. We provide additional discussions the qualitative results and the harmonization quality, supplemented with Fig. 21.
2. Added clarifications on the novelty introduced by our method, please see the highlighted text in the introduction and the methodology along with the detailed algorithm definition provided as Algorithm 1, 2 and 3.
3. Provided further discussions in rebuttal comments regarding the evaluations.
4. Addresses the limitations of our method further, please see Appendix A.2.
5. Fixed notation errors in Section 3.2.
6. Provide clarifications on the benchmark used for evaluation in Sec. 4.2.
7. Address questions regarding the masking coefficient $d$ in our ablation studies provided in section 4.1.4.
8. Address further comparisons with foreground extraction and harmonization methods, regarding the evaluation of our method, where we present comparisons with such methods in Fig. 6 and Fig. 7.
9. Provide further discussions regarding the evaluation setup we use, where we justify the role of each metric in rebuttal comments.

We would be glad to discuss any further concerns about our paper. Thank you again!

Best regards,
Paper 1337 Authors

---

### Meta-Review · Area_Chair_oY5m · 2024-12-26

**Metareview:**

This paper presents an image generation consisting of foreground and background generation branches where the results are blended in the end. While the problem formulation is interesting, however, the reviewers point out that the emphasized harmonization of the foreground and the background to be insufficient. Occasionally, the lighting and the style do not match and shadows could be missing. The final image content could lack generalization as the method is limited to single foreground and there is no physical interaction between the foreground and the background objects. Reviewers generally agree that there is room to improve in this work.

**Additional Comments On Reviewer Discussion:**

A common question from the reviewers was the realisticity of the final generated image. While the blending between the foreground and the background is done at the attention level, the mismatch between the foreground and background was observed in terms of the style, lighting, missing shadows, etc.
The lack of interaction capability between the foreground and the background was also pointed out.
In the discussion, the authors acknowledged several issues.

Besides, several reviewers find the formulation of the image generation pipeline to be unnatural.

---

### Decision · Program_Chairs · 2025-01-22

Reject